# Copper-surface-mediated synthesis of acetylenic carbon-rich nanofibers for active metal-free photocathodes

Tao Zhang[1], Yang Hou[1,2], Volodymyr Dzhagan[3], Zhongquan Liao[4], Guoliang Chai[5], Markus Löffler[6], Davide Olianas[7], Alberto Milani[7], Shunqi Xu[1], Matteo Tommasini [7], Dietrich R.T. Zahn [3], Zhikun Zheng[1], Ehrenfried Zschech[4,6], Rainer Jordan[8] & Xinliang Feng[1]

The engineering of acetylenic carbon-rich nanostructures has great potential in many applications, such as nanoelectronics, chemical sensors, energy storage, and conversion, etc. Here we show the synthesis of acetylenic carbon-rich nanofibers via copper-surface-mediated Glaser polycondensation of 1,3,5-triethynylbenzene on a variety of conducting (e.g., copper, graphite, fluorine-doped tin oxide, and titanium) and non-conducting (e.g., Kapton, glass, and silicon dioxide) substrates. The obtained nanofibers (with optical bandgap of 2.51 eV) exhibit photocatalytic activity in photoelectrochemical cells, yielding saturated cathodic photocurrent of ca. 10 $\mu A\,cm^{-2}$ (0.3–0 V vs. reversible hydrogen electrode). By incorporating thieno[3,2-b]thiophene units into the nanofibers, a redshift (ca. 100 nm) of light absorption edge and twofold of the photocurrent are achieved, rivalling those of state-of-the-art metal-free photocathodes (e.g., graphitic carbon nitride of 0.1–1 $\mu A\,cm^{-2}$). This work highlights the promise of utilizing acetylenic carbon-rich materials as efficient and sustainable photocathodes for water reduction

[1] Center for Advancing Electronics Dresden (cfaed) and Department of Chemistry and Food Chemistry, Dresden University of Technology, Mommsenstrasse 4, 01062 Dresden, Germany. [2] Key Laboratory of Biological Engineering of Ministry of Education, College of Chemical and Biological Engineering, Zhejiang University, Hangzhou 310027, China. [3] Semiconductor Physics, Chemnitz University of Technology, Reichnhainer Strasse 70, 09126 Chemnitz, Germany. [4] Fraunhofer Institute for Ceramic Technologies and Systems (IKTS), Maria-Reiche-Strasse 2, 01109 Dresden, Germany. [5] State Key Laboratory of Structural Chemistry, Fujian Institute of Research on the Structure of Matter, Chinese Academy of Sciences (CAS), Fuzhou 350002, China. [6] Dresden Center for Nanoanalysis (DCN), Dresden University of Technology, Helmholtzstrasse 18, 01069 Dresden, Germany. [7] Dipartimento di Chimica, Materiali ed Ingegneria Chimica 'G. Natta', Politecnico di Milano, Piazza Leonardo da Vinci 32, 20133 Milano, Italy. [8] Chair of Macromolecular Chemistry, School of Science, Dresden University of Technology, Mommsenstrasse 4, 01069 Dresden, Germany. These authors contributed equally: Tao Zhang, Yang Hou. Correspondence and requests for materials should be addressed to X.F. (email: xinliang.feng@tu-dresden.de)

Photoelectrochemical cells (PECs) offer the promise of producing electric energy and hydrogen through artificial photosynthesis by integrating the collection of solar energy and the electrolysis of water into a photoelectrode[1]. PECs are based on photochemical reactions at the junction of semiconductor and electrolyte, in which electrons and holes that generated upon solar absorption by semiconductors (p-type or n-type) are driven into electrolyte solution by applied electric field at the junction, driving a redox reaction, e.g., the reduction of $H^+$ to $H_2$ for p-type semiconductor[2]. To enable their practical use in the field of environmental and clean energy, semiconductor materials need to be low-cost and prepared from abundant resources using scalable approaches[1,3,4], which preclude the utilization of the most reported, efficient PECs systems, such as metal oxides[2,5,6], metal chalcogenides, and transition-metal dichalcogenides[7-10].

Synthetic conjugated polymers, composed of a delocalized π-electron system, present a new generation of sustainable semiconductors for solar-energy utilization[1,3,11-13]. They offer tunable energy levels, low-cost facile synthesis, and respectable solid-state charge-transport characteristics. These promising characteristics have motivated intense investigation into the design and synthesis of conjugated polymer semiconductors for photocatalytic $H_2$ evolution[3,11,12,14,15]. The most representative material is graphitic carbon nitride (g-$C_3N_4$)[11,16-19] and many analogs and composites have also been reported, such as poly(azomethine)s[20], hydrazone-based covalent organic frameworks[21], triazine-based frameworks[22], and biopolymer-activated g-$C_3N_4$[23]. Recently, pyrene-based conjugated polymers have shown promising performance in direct solar water reduction[24-26], suggesting that carbon-rich frameworks are a new family of synthetic polymer semiconductors for solar-to-chemical conversion.

Acetylenic carbon-rich materials (e.g., graphyne, graphdiyne, and related analogs), containing diacetylenic linkages between carbon hexagons in an extended π-conjugation structure, are predicted to exhibit unique electronic, optical, and mechanical properties[27-30]. Recently, the great potential of acetylenic carbon-rich materials as photocatalysts was illustrated by the visible-light-driven degradation of water pollutants (i.e., phenol and methyl orange) using bulk poly(diphenylbutadiyne) nanofibers[29].

In this study, we report an efficient and generic approach for scalable fabrication of acetylenic carbon-rich nanofibers through a Cu-surface mediated Glaser polycondensation. Large-area (up to $4 \times 12$ cm) poly(1,3,5-triethynylbenzene) (PTEB) nanofiber films (with thicknesses from several to hundreds of nanometers) can be grown on various conducting and non-conducting substrates. The obtained PTEB nanofibers are interconnected and have a broad range of visible light absorption (up to 500 nm), corresponding to an optical bandgap of 2.51 eV. We demonstrate that the nanofibers synthesized on conductive substrates can function as metal-free photocathodes for PEC devices, and a saturated photocurrent density up to ca. 10 µA cm$^{-2}$ can be achieved at 0.3–0 V vs. reversible hydrogen electrode (RHE). Furthermore, we show that the PTEB photocathode, by incorporating of thieno[3,2-b]thiophene unit in the conjugated frameworks, exhibited a ca. 100 nm redshift of the absorption edge and a twofold enhancement in the photocurrent, which is superior to those of state-of-the-art metal-free photocathode materials (in the range of 0.1–1 µA cm$^{-2}$; Supplementary Table 1). These results indicate that the Cu-surface-mediated synthetic approach is promising to directly fabricate various acetylenic carbon-rich nanomaterials as photocathodes for PECs.

## Results

### Synthesis and structure characterization.
The impetus for the current synthetic strategy is derived from the observation that metallic copper is able to produce Cu$^{I/II}$ species in polar liquids or alkaline solutions[31,32]. As both Cu$^I$ and Cu$^{II}$ salts have been widely used as catalysts for Glaser coupling reaction[33,34], we expected that the Cu species generated from the metallic copper surface would be able to catalyze the C–C coupling reaction, which could afford acetylenic frameworks on a solid substrate (Supplementary Fig. 1). As illustrated in Fig. 1a, a clean Cu wafer was immersed in a mixture of 1,3,5-triethynylbenzene (TEB, 0.5 mg mL$^{-1}$) and ligand (piperidine, 1 µL mL$^{-1}$) in pyridine at 60 °C for 24 h. The C–C coupling of TEB occurred at the Cu–liquid interface where various Cu species were dissolved, yielding a yellowish PTEB framework deposited directly on the Cu wafer surface (Fig. 1b). In this process, the copper wafer is not only the catalyst (i.e., Cu$^I$ and Cu$^{II}$) source for Glaser coupling reaction but also the substrate for the growth of PTEB nanofibers.

The scanning electron microscopy (SEM) images in Fig. 1c demonstrate that the obtained PTEB nanofibers are distributed uniformly over the entire Cu surface. However, some nanofibers tended to adhere together, resulting in larger bundles ranging from 20 to 80 nm in diameter (Fig. 1c, inset). Individual nanofibers ranging from 5 to 15 nm in diameter could be clearly observed on PTEB (grown on a Cu grid) using transmission electron microscopy (TEM) (Fig. 1d and e). We found that PTEB was grown on the Cu surface at an constant rate of $\delta_d = 10$ nm h$^{-1}$ within 72 h (Supplementary Fig. 2a); therefore, the thickness of the PTEB nanofiber film is controllable in a quite broad range from several (e.g., 6.7 nm) to hundreds of nanometers (e.g., 750 nm) by varying the reaction time (Supplementary Figs. 2b–d). For instance, a 230 nm-thick film of PTEB nanofibers was obtained after 24 h of reaction on the Cu surface (Supplementary Fig. 3a) and further energy dispersive X-ray (EDX) elemental mapping images provide a clear contrast of different layers (PTEB, Cu, and Si) on the cross-section (Supplementary Figs. 3b–g). The film was rather robust and capable of handling and transferring onto arbitrary substrates (e.g., aluminum foil; Supplementary Fig. 4) after etching away Cu with aqueous ammonium persulfate solution (0.1 g mL$^{-1}$) using the standard poly(methyl methacrylate) (PMMA) method (see Methods). The scalability was demonstrated by the preparation of a large-area PTEB film ($4 \times 12$ cm; Supplementary Fig. 5) using only 10 mL of the dilute reaction solution, as indicated above. The resulting nanofiber film was uniform over the entire surface.

Raman spectroscopy has been shown to be one of the best techniques to study the structure of carbon-rich materials and identify diacetylenic moieties[35]. The presence of the Raman line at 2209 cm$^{-1}$ in the PTEB nanofibers (Fig. 1f), assigned to the C≡C stretching, is a strong evidence for the formation of conjugated diacetylenic linkages due to the reaction of the terminal alkyne (which exhibits a C≡C stretch at 2106 cm$^{-1}$; Supplementary Fig. 6)[36,37]. The Raman peaks at 989 and 1581 cm$^{-1}$ can be assigned respectively to the ring breathing and ring stretching of aromatic moieties[38]. The observed peaks match well with the simulated Raman signals obtained from density functional theory (DFT) calculations, which supports the proposed assignments (Methods, Fig. 1f, and Supplementary Figs. 7-9). It should be noted that the slight mismatch of the simulated C≡C stretching peak is expected, partly because of the much higher polymer chain length of the as-prepared PTEB nanofibers (the PTEB model used in DFT calculation)[39], and partly because of intermolecular interactions not included in the model, but operating in the bulk material. X-ray photoelectron spectroscopy (XPS) (Fig. 1g) reveals that the PTEB film contains only elemental carbon. Deconvolution of the C 1 s core level spectra (at 284.1 eV) displays the major fractions of $sp^1$ and $sp^2$ hybridized carbons with binding energies at 283.8 and 284.5 eV, respectively (Fig. 1h)[40-42]. The structure of PTEB was further

confirmed by additional characterizations using Fourier transform infrared spectroscopy (FTIR; Supplementary Fig. 10).

Although we demonstrated that PTEB nanofibers fabricated on Cu are able to be transferred onto arbitrary substrates (Supplementary Fig. 4), the direct growth of nanofibers on a target substrate enables high structural stability and excellent mechanical adhesion, which are required for the fabrication of high-performance devices with long-term stability[43]. It was reported that Cu species (i.e., Cu$^I$ and Cu$^{II}$) generated from a metallic Cu surface could out-diffuse to initiate controlled radical polymerizations on the surface of a facing substrate (Fig. 2a)[31,32]. These results led us to believe that the Glaser coupling reaction, which is catalyzed by similar Cu species, is likely to be achieved

on other substrates with the assistance of a Cu wafer. To this end, a bare and planar substrate (e.g., graphite foil) was sandwiched with a Cu wafer and immersed in the reaction solution as above. After an indicated time interval, the Cu wafer was separated and a uniform PTEB layer was observed on the graphite after thoroughly cleaning with various solvents (e.g., pyridine, dichloromethane, and methanol, sequentially). Finally, a wide variety of substrates, including conducting (e.g., graphite foil, nickel, titanium, Si wafer, and fluorine-doped tin oxide (FTO) glass (Fig. 2b, c and Supplementary Fig. 11a and d) and non-conducting substrates (e.g., Kapton foil, glass, and SiO$_2$ wafer) (Fig. 2d and Supplementary Fig. 11c), were coated with PTEB films by this approach. More interestingly, the morphologies of

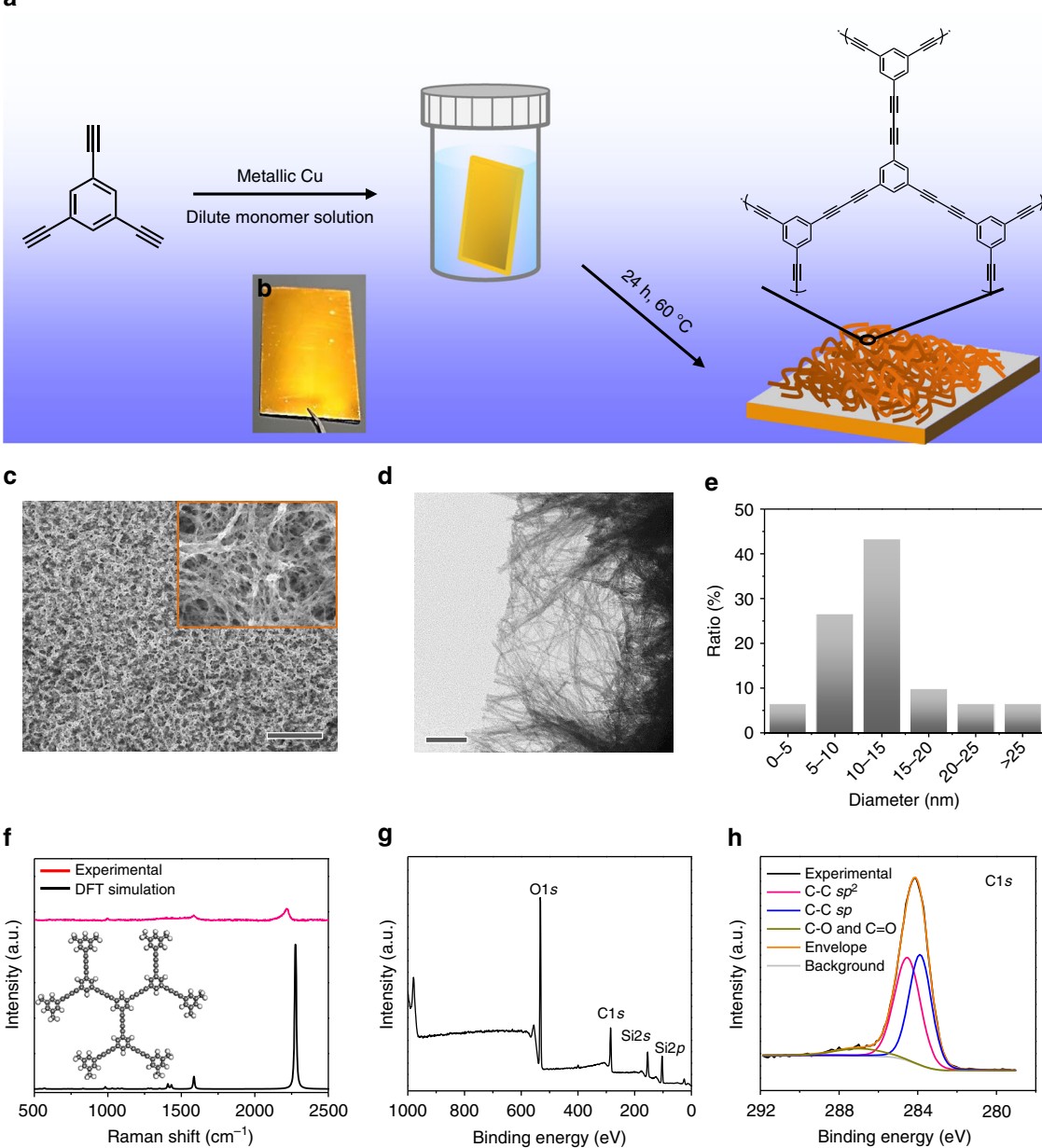

**Fig. 1** Synthesis and characterization of PTEB nanofibers. **a** Reaction scheme and employed molecules. **b** Photograph of the PTEB layer synthesized on a Cu wafer. The coupling reaction occurs only at the Cu–liquid interface, resulting in the selective formation of PTEB netwrok on Cu surface. **c** Scanning electron microscopy (SEM) images of PTEB nanofibers at Cu surface, scale bar: 10 μm. Inset: magnification of **c**. **d** Transmission electron microscopy (TEM) image of PTEB grown on a Cu grid, scale bar: 200 nm. **e** Histogram of the diameter of the PTEB nanofibers measured from **d**. The agreement between the Raman spectra obtained from **f** the experiment and density functional theory (DFT) calculations confirm the generation of diacetylenic linkages in PTEB. **g** XPS survey spectrum and **h** high-resolution C1s core level spectrum of PTEB on a SiO$_2$/Si wafer. The fitting in **h** was performed with a set of Voigt peaks

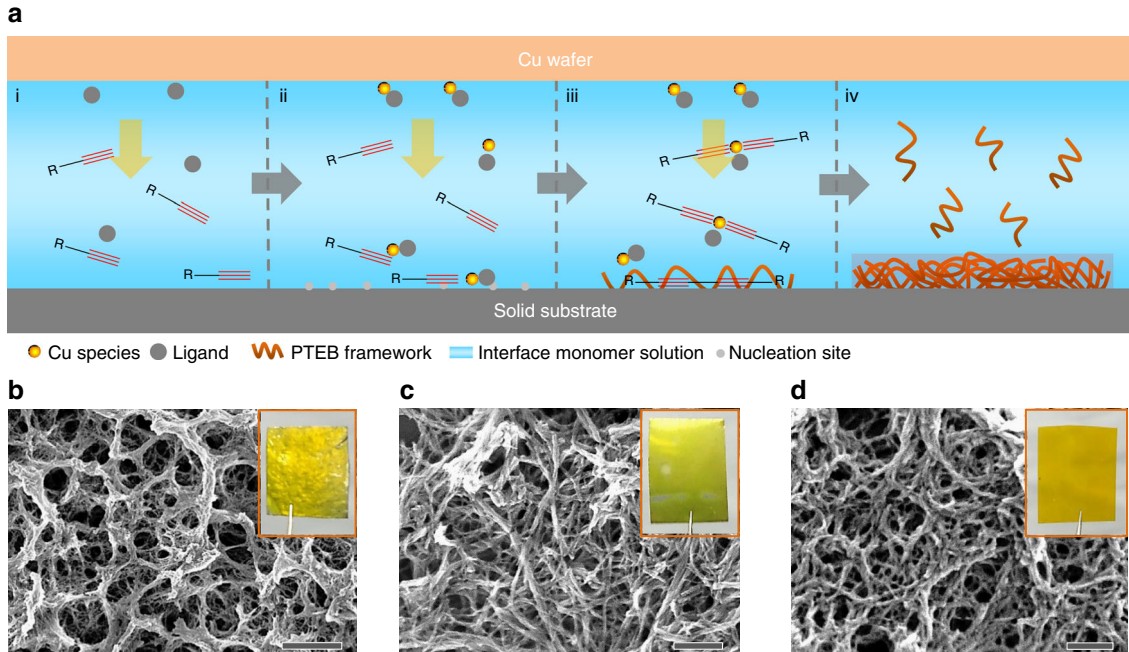

**Fig. 2** Synthesis of PTEB nanofibers on various substrates. **a** Illustration of the synthetic strategy: (i) catalytic Cu species, generated on the surface of the Cu wafer, out-diffuse at the interface with the assistance of a ligand; (ii) the solubilized Cu species react with alkyne terminal monomers (i.e., TEB) and catalyze Glaser polycondensation at the confined interface; (iii) the facing substrate offers nucleation sites to attach oligomers and polymers forming nanofibers; (iv) PTEB nanofibers grow continuously until complete consumption of the monomer at the confined interface. The distance between the Cu wafer and substrate is ca. 0.2 mm. SEM images of the PTEB nanofibers grown on different substrates: **b** graphite foil, **c** nickel plate, and **d** Kapton foil. Insets: photographs of each sample. Scale bar: **b** 1 μm, **c** and **d** 100 nm

the PTEB frameworks vary with the type of substrate. For instance, isolated PTEB nanofibers were observed on the surface of nickel (ca. 8 nm), FTO (ca. 9 nm), Kapton (ca. 10 nm) and glass (ca. 13 nm), whereas the nanofibers obtained on graphite (ca. 40 nm) and titanium surfaces (ca. 15 nm) tended to form large bundles. This result implies that the surface properties of the substrate had a crucial role in the formation of PTEB nanofibers of distinct morphologies, due to heterogeneous nucleation and polymerization process at each substrate (Fig. 2a). Control experiments showed that no such nanofibers were formed when Cu salts (used in classical Glaser coupling) or high concentrations of monomer (i.e., 5 mg mL$^{-1}$) were applied.

**Formation mechanism**. Based on these results, we propose a mechanism according to a previous model describing the synthesis of polyaniline nanofibers on a solid substrate via step-wise electrochemical polymerization[44,45]. Typically, two plausible nucleation sites are proposed, i.e., bulk solution and solid substrate, in the synthesis of PTEB through the Cu-surface-mediated approach. These two nucleation sites compete with each other. For example, concentrated monomer (or catalyst) leads to identical or even faster polymerization of TEB in solution than that of on solid substrates. However, in a dilute monomer (or catalyst) solution, heterogeneous nucleation and polymerization preferably occurs on solid substrate. As such, many reactive nucleation centers can be formed on the solid substrate at a faster rate than in solution at the beginning of the reaction (Fig. 2a). These initial nucleation sites minimize the energy barrier at the interface for the formation of PTEB nanofibers on various substrates. Therefore, the reason for the varied PTEB morphologies on different substrates could be put forward as: different substrates have different surface energy and roughness that resulted in varied nucleation rates, which not only affected the reaction kinetics of

Glaser coupling at the interface, but also the approximation of monomer to the surface of substrate.

Following this scenario, micro-patterned PTEB structures were prepared using a Cu grid closely attached to the substrate as both a catalyst source and a stencil mask (see Methods). Glaser polycondensation can be initiated in the mesh region of the grid, at which the monomer solution and solubilized Cu species are able to interact. In the end, well-defined hexagonal arrays of PTEB frameworks with diameters of ca. 40 μm were fabricated on the SiO$_2$ substrate (Supplementary Figs. 12a–c). Furthermore, the carbon elemental mapping based on EDX spectroscopy confirmed the PTEB patterning (Supplementary Fig. 12d). The corresponding EDX spectra show that the sample contains carbon, oxygen, and silicon, where oxygen and silicon originate from the substrate. Such stencil lithography allows Glaser polycondensation on planar substrates to be spatially patterned over a large area without the use of disruptive materials (e.g., photoresists or chemical etchants), demonstrating the wide potential of this technique[46,47].

**Optical and electronic structure characterization**. The ultraviolet-visible (UV-vis) absorption spectrum of PTEB shows several transition modes in visible wavelengths with an absorption edge of 500 nm (Fig. 3a), and the film has a goldish color on a transparent PET substrate (Fig. 3a, inset). The transmittance spectrum (Supplementary Fig. 13) reveals a ca. 80% transmittance at $\lambda = 550$ nm[48,49]; meanwhile, a maximum ca. 70% absorption is observed for the PTEB layer at $\lambda = 490$ nm, corresponding to an average thickness of ca. 230 nm. We note that there is no obvious difference in the absorption spectra of the PTEB nanofibers grown on different substrates (Supplementary Fig. 14).

The optical bandgap ($E_{bg}$) estimated from the Tauc plot [i.e., plotting $(\alpha h v)^r$ vs. $h v$, where $\alpha$ is absorption coefficient, $h$ Planck constant, $v$ photon frequency, and $r = 2$ for a direct bandgap

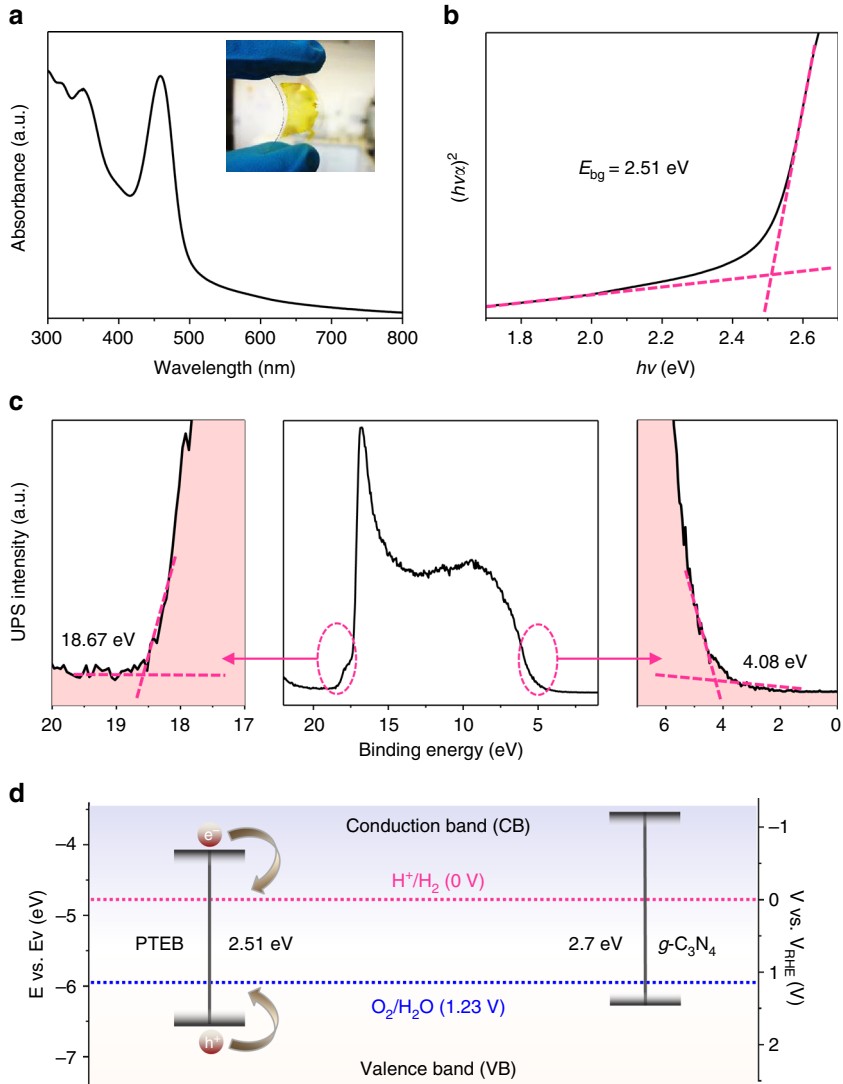

**Fig. 3** Optical and band structure of PTEB nanofibers. **a** UV-vis absorption spectrum. Inset: digital photograph of the PTEB nanofiber film (ca. 230 nm) transferred to a PET substrate. **b** $(h\nu\alpha)^2$ vs. $h\nu$ curve (black curve). The value at the intersection of two dashed red lines of baseline and the tangent of the curve is the bandgap: $E_{bg} = 2.51$ eV. **c** UPS spectrum (black curve). The dashed red lines mark the baseline and the tangents of the curve. The edges of the UPS spectrum are given by the intersections of two dashed red lines of the tangents and the baseline, from which the UPS width is determined. **d** Band structure diagram of the PTEB compared with $g$-$C_3N_4$. In all panels, the average thickness of the PTEB films is *ca.* 230 nm

material] is ~ 2.51 eV (Fig. 3b)[3]. Such bandgap is sufficient to overcome the theoretical endothermic-change in the process of water-splitting (i.e., 1.23 eV). In addition to an appropriate bandgap, the conduction band of a material need to match the donating energy level of water and the valence band match the electron accepting water level that are important to water splitting in PECs. Thus, ultraviolet photoelectron spectroscopy (UPS) was used to determine the energy level of valence band (i.e., $E_{vb}$) of PTEB. The $E_{vb}$ 6.63 eV was calculated by subtracting the UPS width (Fig. 3c) from excitation energy (HeI, 21.22 eV). Furthermore, the conduction band energy $E_{cb}$ was determined to be 4.12 eV from $E_{vb} - E_{bg}$. These values (in vacuum level) were converted to electrochemical potentials according to standard reference electrode, *e.g.*, 0 V vs. RHE is equal to − 4.44 eV vs. vacuum level[3]. We can see from Fig. 3d that the reduction energy level for $H_2O$ to $H_2$ is located below the $E_{cb}$ of PTEB and the oxidation energy level for $H_2O$ to $O_2$ is above the $E_{vb}$ of PTEB, which agrees with DFT calculation results (Supplementary Fig. 15). The electronic band structures of PTEB are consistent with the theory of 1,3,5-graphdiyne reported by Barth et al.[28]

Thus, the proper position of the band structures of PTEB permit the efficient transfer of photo-generated electrons and holes, respectively, and promise the PTEB nanofibers as photoelectrodes for PECs for hydrogen production. Such optical properties are analogous to those of the most representative metal-free photoelectrode material, $g$-$C_3N_4$[11].

**PEC characterization**. PEC experiments were conducted with the PTEB nanofibers on a titanium substrate as photocathode in a solution of 0.01 M $Na_2SO_4$ (pH 6.8) (Fig. 4a). The PEC characterization was performed in a 3-electrode setup with applied bias relative to the reference electrode (Ag/AgCl). The PTEB photocathode gave an apparent photoresponse to light on/off switching at an applied bias under chopped irradiation (100 mW $cm^{-2}$) (Fig. 4b and Supplementary Fig. 16). The appearance of the cathodic photocurrent suggests that PTEB has typical p-type semiconductor behavior[50]. In addition, a saturated cathodic photocurrent density of ca. 10 μA $cm^{-2}$ was obtained on ca. 230 nm-thick PTEB below 0.3 V vs. RHE (after subtracting the

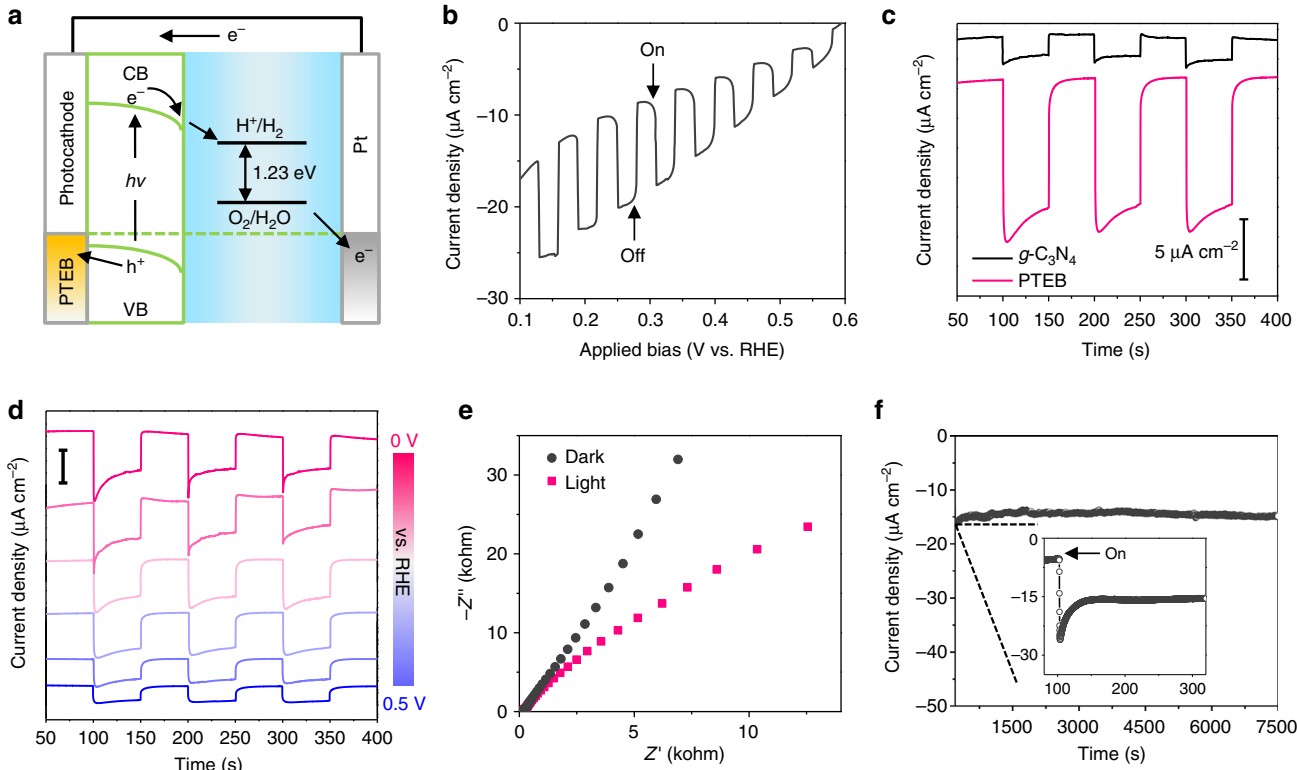

**Fig. 4** PEC characterization of the PTEB nanofiber based photocathodes. **a** PEC cell with a PTEB photocathode under simulated sunlight irradiation (100 mW cm$^{-2}$) in 0.01 M Na$_2$SO$_4$ aqueous solution. **b** Current density-potential curves vs. bias of PTEB under intermittent irradiation. **c** Transient current density vs. time at 0.3 V vs. RHE under intermittent light irradiation for PTEB (red curve) and g-C$_3$N$_4$ (black curve). **d** Photocurrent densities vs. time for PTEB with varied applied potentials (from 0.5 to 0 V vs. RHE) under intermittent irradiation; scale bar: 10 μA cm$^{-2}$. **e** EIS Nyquist plots of PTEB at a voltage of 0.3 V vs. RHE under dark and light irradiation. **f** Current density and efficiencies vs. time of the PTEB electrode under illumination for 7500 s. Inset: magnification of 50–350 s, where light irradiation started at 100 s

dark current). This value is superior to most reported metal-free photoelectrodes, such as g-C$_3$N$_4$[17–19,23,51] and analogs[23,52], red phosphorus[53,54], and silicon carbide[55], which are typically in the range of 0.1–1 μA cm$^{-2}$ (Supplementary Table 1). In our own experiments, the g-C$_3$N$_4$-based photocathode, prepared according to a previously reported method[56], yielded a much lower photocurrent of ca. 2 μA cm$^{-2}$ than the PTEB photocathode (Fig. 4c). We can confirm that the observed photocurrent at PTEB electrodes is due to the light absorption of PTEB nanofibers, as the incident-photon-to-current (IPCE) spectrum matches well with the UV-Vis absorption spectrum (Supplementary Fig. 17). Moreover, we found that the photocurrent strongly related to the thickness of PTEB layer, and both thinner and thicker films gave lower photocurrent density (Supplementary Fig. 18). Therefore, the optimization of PTEB film thickness to reach equilibrium between the light adsorption capability and charge transfer efficiency is necessary to achieve the optimized PEC performance. When a consistent bias voltage of 0.5–0 V vs. RHE (i.e., − 0.1 to − 0.6 V vs. Ag/AgCl) was applied to the PTEB photoelectrode (Fig. 4d), respectively, the transient photocurrents exhibited good switching behavior at all applied bias voltages. A reduced charge-transport resistance under irradiation was detected for the PTEB photocathode, as the arc radius with light irradiation was lower than that in dark (Fig. 4e). These results verify that photoelectrons and holes were generated over the PTEB nanofibers under light irradiation. Given that PTEB is an organic polymer, the stability of the photocathode was further studied. An insignificant change in the photocurrent was observed over 2 h of constant irradiation (Fig. 4f), and the

Raman spectrum remained unchanged after the PEC test (Supplementary Fig. 19), which suggest the good stability of the PTEB photocathodes.

**PTEB-*co*-PDET copolymer photocathode.** For polymeric photocathodes, one of the most important advantages is their rich synthetic modularity, allowing to tailor their structural, optical, and electronic properties. In general, the optical properties of a photocathode material can significantly affect its PEC performance, because photogenerated carriers under light irradiation are prerequisites for PEC reactions[57]. As one example, we show the tuning of the absorption spectrum, and hence the optical bandgap of the PTEB photocathode via the copolymerization of TEB with 2,5-diethynylthieno[3,2-b]thiophene (DET), which has been used extensively in the preparation of high-performance polymers for organic solar cells because of its wide absorption in visible-light region[58]. It can be clearly seen that the incorporation of DET monomer into the PTEB backbone greatly changes its color appearance (i.e., from yellow to red) (Fig. 5a), which results in a > 100 nm redshift of the absorption edge in the UV-vis spectrum (Fig. 5b). The relative loading of DET in the final copolymer structure was determined by EDX spectrum, which gave a structure with subunits PTEB$_{1.3}$-*co*-PDET$_1$ (Supplementary Fig. 20a). Although, SEM image reveals that the PTEB$_{1.3}$-*co*-PDET$_1$ copolymer showing larger nanofiber (bundle) morphology (Supplementary Fig. 20b), we found that the photocurrent density of the copolymer (PTEB$_{1.3}$-*co*-PDET$_1$) was obviously improved (more than twofolds) from ca. 10 μA cm$^{-2}$ to ca. 21 μA

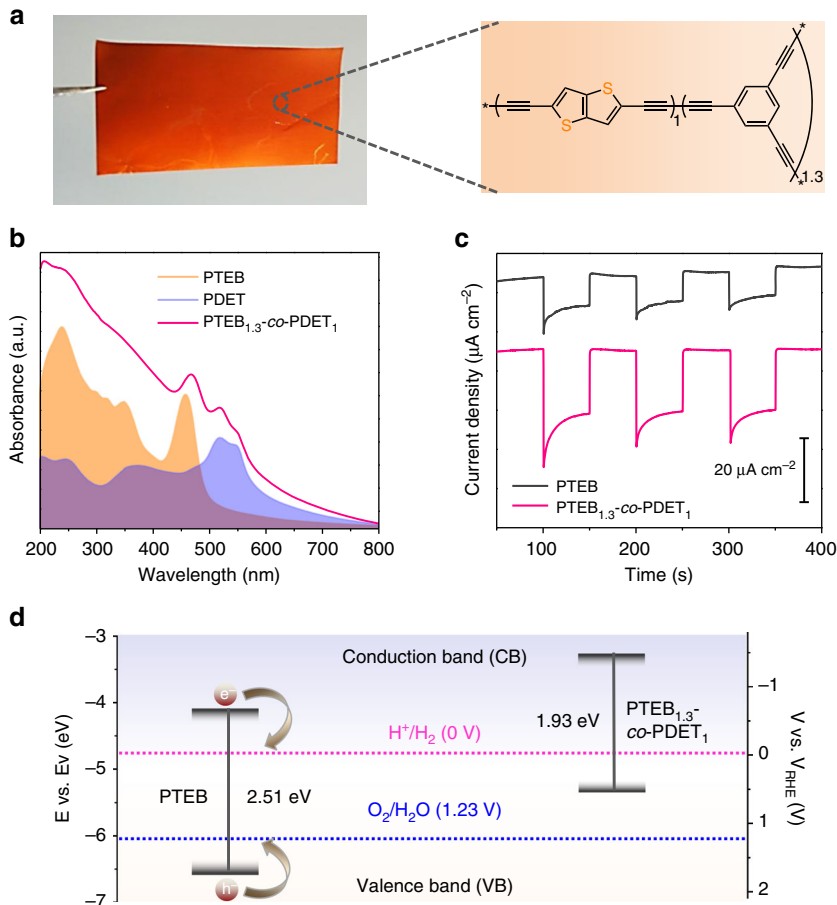

**Fig. 5** PTEB incorporated with poly[2,5-diethynylthieno[3,2-b]thiophene] (PDET) as photocathode. **a** Photograph of $PTEB_{1.3}$-$co$-$PDET_1$ grown on a Cu surface (left) and its chemical structure (right). **b** UV-vis absorption spectra of PTEB, PDET and $PTEB_{1.3}$-$co$-$PDET_1$, respectively. **c** Transient photocurrent density vs. time at a bias of 0 V vs. RHE under intermittent irradiation for $PTEB_{1.3}$-$co$-$PDET_1$ (red curve) and pristine PTEB (black curve). **d** Band structure diagram of the $PTEB_{1.3}$-$co$-$PDET_1$ compared with PTEB. In all panels, the average thickness of the polymer films is ca. 230 nm

$cm^{-2}$ at 0 V vs. RHE (i.e., $-$ 0.6 V vs. Ag/AgCl; Fig. 5c). This value is comparable to that of the $B_{13}C_2$ (ca. 16 $\mu A\ cm^{-2}$ at $-$ 0.76 V vs. Ag/AgCl, i.e., $-$ 0.16 V vs. RHE) (Supplementary Table 1)[50] and even inorganic 2D $WSe_2$ thin films (ca. 40 $\mu A\ cm^{-2}$ at 0 V vs. RHE, i.e., $-$ 0.2 V vs. Ag/AgCl)[9]. The $PTEB_{1.3}$-$co$-$PDET_1$ has a conduction band minimum of $-$ 1.47 V vs. NHE (Fig. 5d and Supplementary Fig. 21), which is much higher than the pristine PTEB ($-$ 0.68 V) as well as theoretical value of $H_2$ evolution (0 V vs. NHE). In addition, the conductivity, $\sigma$, of the $PTEB_{1.3}$-$co$-$PDET_1$ copolymer film was found to be of $1.9 \times 10^{-5}$ $S\ cm^{-1}$, which is an order of magnitude greater than pristine PTEB film (i.e., $3.0 \times 10^{-6}\ S\ cm^{-1}$) (Supplementary Fig. 22). Thereby, it is reasonable to conclude that by introduction of PDET segment in the PTEB structure, not only the light absorption range was enlarged but also the electronic band structure and charge transfer capability for water reduction were improved for the $PTEB_{1.3}$-$co$-$PDET_1$ photocathode.

## Discussion

To gain insights on the PEC activity of PTEB nanofiber for water reduction, we evaluated the effects of the sacrificial reagents on photocurrent density of PTEB photocathode. In this respect, the transient photocurrent density of PTEB photocathode was examined in 0.01 M $Na_2SO_4$ electrolyte (pH 6.8) in the presence of electron scavenger ($10^{-5}$ M $Cu^{2+}$, it reacts with electron to yield $Cu^+$)[59]. The reduction of $Cu^{2+}$ is thermodynamically and

kinetically more facile than the reduction of water. As such, the cathodic photocurrent density at PTEB nanofibers is noticeably accelerated (to ca. 18 $\mu A\ cm^{-2}$ at 0.3 V vs. RHE) in the presence of $Cu^{2+}$ (Supplementary Fig. 23). The significant enhancement of photocurrent implies that the electron-hole recombination rate decreased due to the reaction of $Cu^{2+}$ with photogenerated electron, leaving excess holes at the photoelectrode. We analyzed the gaseous product from the PEC cell after irradiation using a gas chromatograph (GC) and a moderate amount of $H_2$ production (2.53 $\mu mol$ in 10 h at 0 V vs. RHE) was detected on PTEB cathode (Supplementary Fig. 24). The amount of $H_2$ production is close to the value from theoretical calculation, suggesting the photocurrent of PTEB cathode mainly attributing to the PEC water reduction (Supplementary Fig. 24). Furthermore, in a PTEB nanofibers film-based photocatalytic cell[60,61], a total amount of 11.4 $\mu mol$ $H_2$ gas was produced after 10 h reaction without noticeable deterioration of the activity within 30 h (Supplementary Fig. 25). The average $H_2$ evolution rate of the PTEB nanofibers was about 1.14 $\mu mol\ h^{-1}$, with an apparent quantum efficiency of 1.83% at 420 nm (see Methods). Notably, a particularly high rate of > 11,400 $\mu mol\ h^{-1}\ g^{-1}$ for photocatalytic $H_2$ evolution was obtained, if the mass weight (< 0.1 mg) of PTEB nanofibers film was considered. To gain more insights on the active sites of PTEB for $H_2$ evolution reaction, the reaction process of proton adsorption–reduction–hydrogen adsorption was simulated using DFT calculations[62] and the free-energy changes were calculated regarding to four different carbon atoms of PTEB

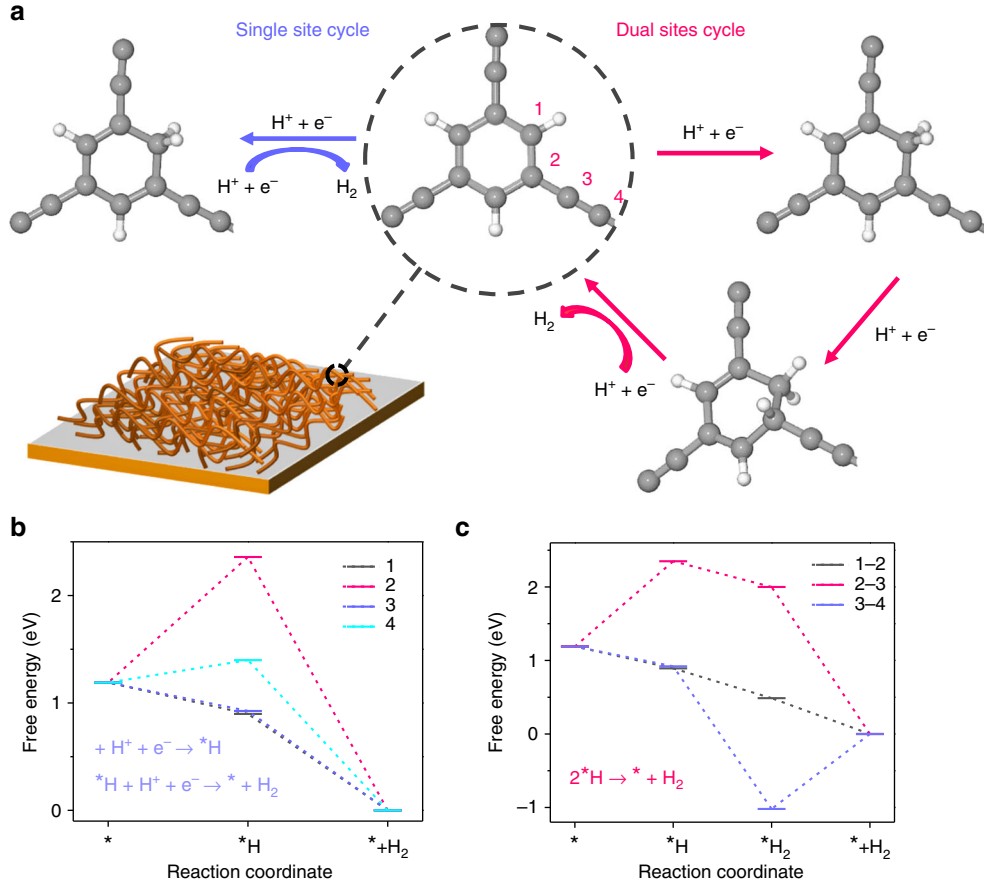

**Fig. 6** DFT calculation to investigate the $H_2$ evolution active sites. **a** Reaction cycles and active sites for single and dual sites $H_2$ evolution from water. **b** Free-energy variations for $H_2$ evolution via single site reaction pathway: 1, 2, 3, and 4 denote for different active sites as labeled in **a**. **c** Free-energy variations for $H_2$ evolution via dual sites reaction pathway: sites 1–2, 2–3, and 3–4 denote for different active sites as labelled in **a**. *Catalyst (i.e., PTEB) surface

(Fig. 6a). The reaction pathways for both single and dual sites $H_2$ evolution from water reduction were studied, and corresponding free-energy variations indicate that site 1 and site 3 are favorable for single-site $H_2$ evolution (Fig. 6b), and the sites 1 and 2 are favorable for dual-site $H_2$ evolution (Fig. 6c). The DFT results imply that carbon atoms of benzene ring (in PTEB) are dominant active sites for photocatalytic $H_2$ evolution, which agrees with the results from Cooper et al.[24] proving that carbon-rich polymers (based on phenylenes and pyrenes) are able to catalyse photocatalytic $H_2$ production from water.

In this work, the superior PEC performance of the PTEB nanofibers can be shown in the following ways: first, the light adsorption region of a photoelectrode can significantly affect its usable light source. The PTEB nanofibers absorb light in a wide visible range from ultraviolet to blue region and show a similar profile as $g$-$C_3N_4$ (bandgap of 2.7 eV)[11] with an absorption edge around 500 nm (bandgap of 2.51 eV) (Fig. 3a, b). This indicates that PTEB can be excited by broader regions of solar light and a large amount of electron and holes can be produced under irradiation. Second, the efficiency of electron-hole separation is another crucial factor that determines the performance of a photoelectrode. The interconnected nanofibrous structure of PTEB offers a short diffusion distance, which results in an enhanced charge transport and a high surface area for fast interfacial charge collection, largely contributing to the considerable PEC activity (Fig. 2b–d and Supplementary Figs. 11 and 26)[43]. More importantly, the direct growth of PTEB frameworks on conductive substrates can greatly enhance electron transfer and adhesion between the substrate and the active component and enhance the structural stability for long-term operation[29,43].

In conclusion, the method described herein affords a facile and scalable approach for the synthesis of acetylenic carbon-rich nanofibers via Cu-surface mediated Glaser polycondensation. In this process, both conductive and non-conductive substrates can be uniformly coated with PTEB nanofibers; meanwhile, micropatterned PTEB was achieved using a patterned Cu grid as a stencil mask. We demonstrated that the PTEB nanofibers fabricated on conductive substrates can be directly utilized as metal-free photocathodes in PEC for $H_2$ production, affording a saturated photocurrent of ca. 10 μA cm$^{-2}$ at 0.3–0 V vs. RHE. The achieved photocurrent is largely improved to ca. 21 μA cm$^{-2}$ through the introduction of thieno[3,2-b]thiophene units in the backbone of PTEB framework. These results clearly illustrate that acetylenic PTEB frameworks can serve as a promising polymeric photocathode in PEC devices for hydrogen production. Owing to the diversity of terminal alkynes and the chemical tailorability of the C≡C triple bond (e.g., thiol-yne reaction[63], cycloaddition with cyano-containing acceptor molecules[64], and metal coordination[65]), it is feasible to further improve the PEC performance with a much broader set of acetylenic carbon-rich frameworks and composites. Therefore, this work offers opportunities in the development of metal-free photocathode materials for solar water reduction.

## Methods

**Materials.** All the reagents were obtained from Sigma-Aldrich and used as received. Copper wafer (MicroChemicals GmbH, Germany): Prime CZ-Si wafer 4 inch, one side polished, p-type (boron), total-thickness-variation < 10 μm, 1–10 Ω cm; 10 nm titanium adhesion layer; 200 nm copper (purity > 99.9 %), root-mean-square roughness < 10 nm. The copper foil (thickness 0.25 mm, 99.98%) was purchased from Sigma-Aldrich. The copper was consecutively washed with portions of 3 M HCl (in methanol), methanol and ethanol under ultrasonication (2 min), and dried under a flow of argon. The cleaned copper wafer was immediately used for catalysis.

**Synthesis of PTEB on Cu wafer or foil.** Typically, TEB (5 mg, 0.033 mmol) and piperidine (10 μL, 0.1 mmol) were added in a glass bottle containing 10 mL pyridine as solvent. The freshly cleaned copper was submerged into a reaction mixture. Afterwards, the bottle was sealed and heated to 60 °C in an oven for a certain time. Repetitive series of experiments gave no significant differences in terms of the resulting PTEB layer thickness and morphology. After reaction, the samples were immediately washed with fresh pyridine, dichloromethane, and methanol sequentially. Finally, the substrates were blow-dried by a jet of dry nitrogen and a golden yellow film was obtained uniformly on the substrate.

**Transfer of PTEB.** To transfer the PTEB film on copper to another substrate (e.g., PET), the film was coated with PMMA resist (Allresist GmbH product number AR-P671.04, dissolved in chlorobenzene), and cured at 90 °C for 10 min. The copper substrate was etched away by a water solution of ammonium persulfate (0.25 g mL⁻¹) in 2 h. After being rinsed thoroughly with deionized water, the PMMA/PTEB film was transferred to a target substrate. The samples were naturally dried in air for 1 h and stored in high vacuum (room temperature) for 24 h to enhance the adhesion of PTEB with targeted substrate surface. PMMA was removed by thorough rinsing in acetone and cured in isopropyl alcohol.

**Synthesis of PTEB on other substrates.** A planar substrate (e.g., SiO₂ wafer, graphite, titanium, nickel, FTO, glass, and Kapton) piece cleaned by water and ethanol was sandwiched with a copper wafer in a distance of $d = 0.1$ mm adjusted by two spacers. The assembly was immersed in the reaction mixture as indicated above and the washing procedures are similar.

**Patterned PTEB network.** Patterned PTEB film was fabricated on SiO₂ wafer by using copper grid as both a catalyst source and a stencil mask. The samples were clamped with copper TEM grids with various hole sizes (Plano, Germany) and immersed in the reaction mixture as described above. The distance between the copper grid and substrate has an important role on fabricating positive and negative patterns on the substrate, where negative patterned PTEB film was obtained by a direct attachment of copper grid to the substrate and positive pattern was achieved by close attachment assisted by the evaporation of a drop of iso-propanol due to capillary force.

**Synthesis of g-C₃N₄.** Bulk g-C₃N₄ was synthesized according to a reported procedure with some modifications. In a typical synthesis, 5.0 g urea was heated at 550 °C in Ar atmosphere with a rate of 2 °C per min for 4 h[56]. The obtained sample was deposited onto as-washed (using 3 M HCl in methanol) titanium plate with controlled thickness of ca. 250 nm using spin-coating to form a film photoelectrode.

**Synthesis of PTEB₁.₃-co-PDET₁ on Cu substrate.** Typically, TEB (2.5 mg, 0.017 mmol), DET (2.5 mg, 0.013 mmol), and piperidine (8.9 μL, 0.09 mmol) were added in a glass bottle containing 10 mL pyridine as solvent. The freshly cleaned copper was immersed in the reaction mixture. The following procedures are similar to the synthesis of PTEB on Cu substrate.

**Synthesis of PTEB₁.₃-co-PDET₁ on other substrates.** A planar substrate (e.g., titanium and quartz glass) piece cleaned by water and ethanol was sandwiched with a copper wafer in a distance of $d = 0.1$ mm adjusted by two spacers. The assembly was immersed in the reaction mixture mentioned above and washing procedures are similar.

**PEC measurements.** The polarization curves of as-prepared PTEB nanostructured on titanium plate as photocathodes were performed using a three-electrode setup contains working electrode (PTEBs), counter electrode (Pt wire), and reference electrode (Ag/AgCl). The simulated sunlight was from a 200 W Xenon lamp (100 mW cm⁻²) coupled with an AM 1.5 G filter (Newport). An electrochemical analyzer (CHI 760 E) was applied to measure the LSV characteristic of the electrodes, with 1 mV s⁻¹ scan rate, and there is no correction on data for any losses of uncompensated resistance. The electrolyte (0.01 M Na₂SO₄, pH = 6.8) was degassed for 30 min by flushing high purity argon at room temperature (ca. 25 °C) before the measurement. The EIS spectra were recorded by applying a 10 mV AC signal in the frequency range from 100 K to 0.01 Hz at a DC bias of 0.3 V vs. RHE (i.e. −

0.3 V vs. Ag/AgCl). Current density was calculated using the exposed geometric surface area of 1.0 cm² of the photoelectrode

$$(J_{photocurrent\ density} = J_{measured\ photocurrent} / S_{exposed\ geometric\ surface\ area}) \qquad (1)$$

The applied potential vs. Ag/AgCl is converted to RHE potential using the following equation:

$$E_{RHE} = E_{Ag/AgCl} + 0.059 pH + E^0_{Ag/AgCl} (E^0_{Ag/AgCl} = 0.199V) \qquad (2)$$

The amount of H₂ evolved on the PTEB photocathode was measured by GC equipped with a thermal conductivity detector (TCD, N₂ carrier) at 0 V vs. RHE of applied bias in 0.01 M Na₂SO₄ solution under AM 1.5 G irradiation (100 mW cm⁻²)

The IPCE was measured by using a Xenon lamp (100 mW cm⁻², AM 1.5 G) with specific wavelength filters to select the required wavelength of light. IPCE can be expressed as:

$$IPCE(\%) = \frac{J_{ph}(mA/cm^2) \times 1240(V \cdot nm)}{\lambda(nm) \times J_{light}(mW/cm^2)} \times 100 \qquad (3)$$

where the $J_{Ph}$ and $J_{light}$ are the real photocurrent density and light intensity at the wavelength $\lambda$.

**Photocatalytic H₂ evolution.** The PTEB nanofibers film-based photocatalytic cell for H₂ evolution test was constructed according to a reported process[60,61]. In brief, the PTEB nanofibers film (3 × 3 cm², ca. 230 nm-thick) was placed in the center of a gas-closed reaction cell with 120 mL 25% triethanolamine water solution with magnetic stirring. The temperature of the reaction system was kept at around 25 °C. A 200 W Xenon lamp with a filter of $\lambda > 420$ nm was applied to execute the photocatalytic reaction. The amount of H₂ produced from water was determined by GC equipped with a TCD.

The monochromatic illumination quantum yield (MIQY) for H₂ evolution was characterized using a similar setup but with a 420 nm band-pass filter. The MIQY was calculated based on the equation:

$$MIQY(\%) = \begin{aligned} & 100 \times 2 \\ & \times (\text{the number of evolved } H_2 \text{ molecules}) \\ & / \text{the number of incident photos} \end{aligned} \qquad (4)$$

The number of the incident photons was determined using a radiant power energy meter (Newport). The produced H₂ molecules reached 1.3 μmol in 10 h, and the MIQY was calculated as 1.83 %.

**DFT calculation.** Raman spectra: calculations were performed at B3LYP/6-31 G(d,p) level of theory with the Gaussian09 suite of programs[66] on a cluster model representing a section of PTEB (Supplementary Fig. 7). The equilibrium structure of this model, its Hessian, and polarizability derivatives were used as input to compute the simulated Raman spectrum (Fig. 1f) in presence of heavy mass (100 amu) at the peripheral atoms. As confirmed by a comparison (Supplementary Fig. 8) with the results from periodic boundary conditions calculation (Crystal14[67], same functional and basis set as above), this approach allows to effectively quench the vibrations of the peripheral moieties of the cluster model and their contributions to the simulated Raman spectrum. However, the results from the Gaussian09 calculation allow to more straightforwardly analyse and assign the vibrational modes with a suite of ad hoc programs developed for graphene molecules in Milano (Supplementary Fig. 9).

Hydrogen evolution reaction: DFT calculations were carried out by using quantum ESPRESSO code[68]. The generalized gradient approximation of Perdew–Burke–Ernzerhof was used for exchange correlation functional in DFT[69]. In all the cases, spin polarization was considered in the calculation. The kinetic energy cutoffs were set to 35 Ry and 350 Ry, respectively, for the wavefunction and the charge. 3 × 3 × 1 k-points grids were used for structure optimization and total energy calculations. The adsorption energies of intermediates are calculated by using H₂O (l) and H₂ (g) as references. The free-energy variation is obtained by DFT total energy calculations through adding corrections to entropy, zero point energy, and solvation energy according to the method developed by Nørskov et al.[70]. Thus, free energies can be obtained from total energies of intermediates by adding some corrections:

$$\Delta G = \Delta E_{Total} + \Delta E_{ZEP} - T \Delta S + \Delta G_s \pm 0.0592 pH \pm eU(S1) \qquad (5)$$

where $E_{Total}$ is DFT calculated total energy, $\Delta E_{ZPE}$ zero point energy, $\Delta S$ entropy, and $\Delta G_s$ (– 0.22 eV) solvation energy for reaction intermediate. The pH effect were considered as 0.0592 pH and – 0.0592 pH for hydrogen evolution reaction (HER). We set pH 7 in all the calculations. By considering external potential $U$, the free energies corrected by $eU_{red}$ and $– eU_{ox}$ for HER. According to the band structure in the main text, the $U_{red}$ and $U_{ox}$ for PTEB equal to $– 1.01$ and $1.40$ V, respectively. Both single site (Volmer–Heyrovsky) and dual sites (Volmer–Tafel) reaction pathways were investigated for HER. The elementary steps for single-site HER

process are:

$$* + H^+ + e^- \rightarrow {}^* H \tag{6}$$

$$^*H + H^+ + e^- \rightarrow {}^* + H_2 \tag{7}$$

where * denotes catalysts surface. For dual sites HER, the first and second elementary steps are of electron transfer steps that are like Eq. 6, and the third step is a Tafel step:

$$2^*H \rightarrow {}^* + H_2 \tag{8}$$

**Data availability**. The data that support the findings of this study are available from the corresponding author on request.

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

# ARTICLE

52. Chana, D. K. L. & Yu, J. C. Facile synthesis of carbon- and oxygen-rich graphitic carbon nitride with enhanced visible-light photocatalytic activity. *Catal. Today*, https://doi.org/10.1016/j.cattod.2017.05.017 (2017).

53. Wang, F. et al. Red phosphorus: an elemental photocatalyst for hydrogen formation from water. *Appl. Catal. B Environ.* **111**, 409–414 (2012).

54. Hu, Z. F., Yuan, L. Y., Liu, Z. F., Shen, Z. R. & Yu, J. C. An elemental phosphorus photocatalyst with a record high hydrogen evolution efficiency. *Angew. Chem. Int. Ed.* **55**, 9579–9584 (2016).

55. Wang, B. et al. Mesoporous silicon carbide nanofibers with in situ embedded carbon for co-catalyst free photocatalytic hydrogen production. *Nano Res.* **9**, 886–898 (2016).

56. Hou, Y., Wen, Z. H., Cui, S. M., Feng, X. L. & Chen, J. H. Strongly coupled ternary hybrid aerogels of N-deficient porous graphitic-$C_3N_4$ nanosheets/N-doped graphene/NiFe-layered double hydroxide for solar-driven photoelectrochemical water oxidation. *Nano Lett.* **16**, 2268–2277 (2016).

57. Bao, J. M. Photoelectrochemical water splitting a new use for bandgap engineering. *Nat. Nanotechnol.* **10**, 19–20 (2015).

58. Bronstein, H. et al. Thieno[3,2-b]thiophene-diketopyrrolopyrrole containing polymers for inverted solar cells devices with high short circuit currents. *Adv. Funct. Mater.* **23**, 5647–5654 (2013).

59. Ghosh, S., Priyam, A., Bhattacharya, S. C. & Saha, A. Mechanistic aspects of quantum dot based probing of Cu (II) ions: role of dendrimer in sensor efficiency. *J. Fluoresc.* **19**, 723–731 (2009).

60. Wang, Q. et al. Particulate photocatalyst sheets based on carbon conductor layer for efficient z-scheme pure-water splitting at ambient pressure. *J. Am. Chem. Soc.* **139**, 1675–1683 (2017).

61. Wang, Q. et al. Scalable water splitting on particulate photocatalyst sheets with a solar-to-hydrogen energy conversion efficiency exceeding 1%. *Nat. Mater.* **15**, 611–615 (2016).

62. Chai, G. L. et al. Active sites and mechanisms for oxygen reduction reaction on nitrogen-doped carbon alloy catalysts: Stone-Wales defect and curvature effect. *J. Am. Chem. Soc.* **136**, 13629–13640 (2014).

63. Enriquez, A. et al. Efficient thiol-yne click chemistry of redox-active ethynylferrocene. *Organometallics* **33**, 7307–7317 (2014).

64. Li, Y. R., Ashizawa, M., Uchida, S. & Michinobu, T. Colorimetric sensing of cations and anions by clicked polystyrenes bearing side chain donor-acceptor chromophores. *Polym. Chem.* **3**, 1996–2005 (2012).

65. Yam, V. W. W., Hui, C. K., Yu, S. Y. & Zhu, N. Y. Syntheses, luminescence behavior, and assembly reaction of tetraalkynylplatinate(II) complexes: crystal structures of $[Pt(^tBu_3trpy)(C{:}CC_5H_4N)Pt(^tBu_3trpy)]$ $(PF_6)_3$ and $[Pt_2Ag_4(C{:}CC{:}CC_6H_4CH_3\text{-}4)_8(THF)_4]$. *Inorg. Chem.* **43**, 812–821 (2004).

66. Frisch, M. J. et al. Gaussian 09 (revision D.01) (Gaussian, Inc., Wallingford CT, 2016).

67. Dovesi, R. et al. CRYSTAL14: a program for the ab initio investigation of crystalline solids. *Int. J. Quantum Chem.* **114**, 1287–1317 (2014).

68. Paolo, G. et al. QUANTUM ESPRESSO: a modular and open-source software project for quantum simulations of materials. *J. Phys. Condens. Matter* **21**, 395502 (2009).

69. Perdew, J. P., Burke, K. & Ernzerhof, M. Generalized gradient approximation made simple. *Phys. Rev. Lett.* **77**, 3865 (1996).

70. Nørskov, J. K., Rossmeisl, J., Logadottir, A. & Lindqvist, L. Origin of the overpotential for oxygen reduction at a fuel-cell cathode. *J. Phys. Chem. B* **108**, 17886–17892 (2004).

## Acknowledgements

This work was financially supported by the ERC Grant 2DMATER, ESF Young Researcher Group 'GRAPHD,' and the EC under the Graphene Flagship (number CNECTICT-604391). The German Excellence Initiative via the Cluster of Excellence EXC1056 "Center for Advancing Electronics Dresden" (cfaed) is gratefully acknowledged. We gratefully acknowledge Dr. Renhao Dong for helpful discussion and Kejun Liu for assistance in the characterization of SEM. D.O., A.M., and M.T. acknowledge funding support from the European Research Council (ERC) under the European Union's Horizon 2020 research and innovation programme ERC–Consolidator Grant (ERC CoG 2016 EspLORE grant agreement No 724610).

## Author contributions

T.Z. and X.F. conceived and designed the experiments and wrote the paper. T.Z. carried out most of the experiments. Y.H. designed and performed PEC. V.D. and D.R.T.Z. measured and analysed XPS and UPS. Z.L. and E.Z. assisted with the TEM characterization. G.C. performed DFT calculations of band structure and $H_2$ evolution reactions. M.L. measured SEM. D.O., A.M., and M.T. carried out DFT calculations of Raman spectrum and related data analysis. S.X. prepared the DET monomer. Z.Z. and R.J. rendered helpful discussions. All authors discussed the results and commented on the manuscript.

## Additional information

**Competing interests:** The authors declare no competing interests.

