## [Peer Review File(PDF 1068 kb) · Nature Communications]

Reviewers' comments:

Reviewer #1 (Remarks to the Author):

Photo-induced water splitting is regarded as promising and attractive pathway for achieving solar-to-fuel conversion, and effective hydrogen evolution using earth-abundant and low-cost elements is with vital importance. In this manuscript, the authors have established an on-surface synthesis approach for the production of acetylenic carbon-rich nanofibers via the Cu-mediated Glaser polycondensation of 1,3,5-triethynylbenzene (TEB) on a variety of substrates for PEC hydrogen evolution. The result presented here is interesting. However, we still cannot recommend the publication of this manuscript in Nature Communication at this stage for the following reasons: 1) the production of H₂ has not been confirmed; 2) the mechanism of proton reduction is not provided; 3) the role of introduced DET unit is not very clear; the efficiency of this system is not astonishing; and 4) finally, the manuscript is finished in a hurry, lots of mistakes can be observed. The specific comments are below:

1. According to the reported results (J. Phys. Chem. C 2012, 116, 9398–9404), the potentials vs Ag/AgCl can be converted to the reversible hydrogen electrode (RHE). In order to convert a bias vs. Ag/AgCl into vs. RHE, the temperature and pH value of the solution should be provided.
2. In Fig. 4c, PTEB nanofiber based photocathode can give a photocurrent of $\sim 10 \mu\text{A cm}^{-2}$ at a bias of $-0.3 \text{ V vs. Ag/AgCl}$. However, the photocurrent density still keeps at $\sim 10 \mu\text{A cm}^{-2}$ when the external bias increased to $-0.6 \text{ V vs. Ag/AgCl}$ (Fig. 5c), why? For a better comparison, the photocurrent responses at -0.4 , -0.5 and $-0.6 \text{ V vs. Ag/AgCl}$ should be added in Fig. 4d.
3. The authors claim that electrons accumulated in PTEB are consumed to form H₂ by reducing water. So, my question is, have you directly detected H₂ gas using gas chromatography? If yes, please give us the corresponding IPCE value of this system? Also, if H₂ gas does evolve here, could you explain us the active sites or mechanism of proton reduction on the metal-free material?
4. In order to directly confirm the proton reduction property of PTEB, we think that photocatalytic hydrogen evolution performance of this material can be evaluated in the presence of sacrificial reagents, such as ascorbic acid, methanol, or triethylamine etc..
5. The resolution/magnification of TEM image in Fig. 1d is too low to give the corresponding size distribution of PTEB in Fig. 1e. We do think that TEM images with a much higher resolution are necessary.
6. In Fig. 1f, the author assigned the peaks at 989 cm^{-1} and 1581 cm^{-1} respectively to the ring breathing and ring stretching modes of the aromatic moieties. We think that proper references should be cited here to support your claim.
7. It can be observed that the morphology of the obtained PTEB on different substrates is different. So, what's the reason? And could you control the morphology of PTEB through a facile pathway? Besides, have you explored the influence of morphology on PEC performance?
8. In page 9, the authors claim that "we note that there is no obvious difference in the absorption spectra of the PTEB nanofibers grown on different substrates." Please provide us the UV-vis absorption spectra of PTEB grown on different substrates.
9. Do the authors think that the thickness of PTEB would exert an influence on PEC performance? You'd better provide us direct evidences.

10. The PEC performance of PTEB can be further improved by introducing DET unit into the backbone. We think that the author must explain us the intrinsic reasons of the enhanced PEC performance. Is it merely a matter of enhanced light absorption? Will the band gap, CB and VB, change either? We also think that the conductivity of PTEB before and after DET introduction should be compared.

11. In Supplementary Figure 2a, the unit of x-axis should be hour (h). In Supplementary Figure 2b, we think that the the unit of x-axis should be micron (μm).

12. In Supplementary Figure 3, a cross-section SEM image of PTEB on a copper coated Si wafer was provided. We think that, for a better view of the structure, corresponding elemental mapping should be provided.

13. This manuscript is finished in a hurry. As a result, there are many style mistakes. For example, the unit of time (second) should be written in s, not S (Figure 4); and the first letter should be capitalized of each sentence (see corresponding mistakes in the caption of every figure).

Reviewer #2 (Remarks to the Author):

The paper describes the preparation, characterization and photoelectrochemical response of a polybutadiene polymer deposited onto metallic and carbon substrates. The material is prepared by a Cu-catalyzed reaction (Glaser) between terminal aryl acetylenes. The key finding reported is that the modified electrodes exhibit a cathodic photocurrent arising from reduction of protons (however production of H₂ is not confirmed experimentally). The paper is reasonably well-written, and the work is novel. While the paper does contain somewhat exaggerated claims (see below), the work should be of sufficient interest to warrant publication in the journal. I recommend publication pending revision to address a number of significant comments as outlined below.

1) Exaggerated/unsubstantiated claims:

a) A photocurrent of 10 μA under AM1.5 illumination corresponds to <0.1% overall quantum efficiency. Given this, I recommend that the authors remove all hyperbole such as "highly active" (title), "excellent photocatalytic activity" (abstract), "excellent charge transport" (unsubstantiated by any conductivity or mobility measurements), "excellent polymeric photocathode..", etc.

b) Water-splitting has not been demonstrated. At best, proton-reduction is accomplished, but even this is not proved. I strongly recommend that evidence for H₂ production is provided before the paper can be accepted. Even if H₂ production is demonstrated, I recommend that the use of "water splitting" is not used in the paper to describe what amounts to a photocathodic reaction. Water splitting implies that the material is able to carry out the reaction, H₂O \rightarrow H₂ + O₂, and while O₂ may be generated at a counter electrode, this is under the (unknown) applied bias of a 3-electrode configuration using a potentiostat, and the actual potential applied to the auxilliary electrode is unknown.

2) Absolute monochromatic illumination quantum yield and IPCE spectrum should be reported. This will highlight the true potential for this and related materials to be useful in solar energy conversion.

3) The pH of the electrolyte solution is not reported, and this is quite an important parameter.

4) Throughout the paper, the authors imply that the polybutadiene has a "high conjugation length"

(text, p. 6). This is not at all likely. The backbone structure consists of phenyl-CC-CC-phenyl units linked through meta-connections. This type of structure is not very strongly delocalized (see for example, DOI: 10.1021/ja029489h). Singlet excitons in such meta-linked phenyl-ethylene structures are confined to very short segments, and if there is much delocalization between multiple chains (interchain excitons), these also will be spatially confined. The same can be said for the charged states of the material (polarons). Thus, transport in the material will be relatively poor, because the dominant mechanism is hopping. Therefore, the authors should restrict their use of exaggerated claims about "excellent transport", etc. Indeed, the EIS plot in Figure 4e suggests that the material is resistive (e.g., poor transport).

5) The Glaser reaction is catalyzed by Cu(I), and the authors are using metallic Cu (copper(0)). Presumably dioxygen is the oxidant? The authors should make this clear.

6) p. 10, value used to convert from vacuum level scale to NHE (-4.44 V) is not correct. This means the redox potentials shown for the material in Figure 3d are not correct. The oxidation and reduction potentials should be shifted negative by at least 400 mV, see: 10.1002/adma.201004554. This does not change the major conclusion that the polymer has the thermodynamic potential to split water, but the values reported in the diagram are incorrect.

7) Minor comments:

a) Labels a, b, c in the figures are too small and are difficult to discern, especially in Figure 1. All instances should be enlarged.

b) Binding is misspelled in x-axis in Figure 1 (2 instances)

c) planar is misspelled on p. 6 (planer)

d) What is the value alpha (absorption coefficient) for the material (equation on p. 10)?

e) Text should make it clear that the photoelectrochemistry is carried out in a 3-electrode cell with applied bias relative to the reference electrode (not the counter/auxilliary electrode).

f) p. 11, strike "Remarkably"; as noted above 10 uA is hardly remarkable.

e) Figure 5: molecular structure for the copolymer is misleading. It is not known whether this is a truly alternating co-polymer structure. The relative loading of the co-monomer is not determined. The structure should show the subunits within parenthesis () with x and y subscripts.

Reviewer #3 (Remarks to the Author):

The manuscript by Tao Zhang et al reports the Cu-mediated Glaser-coupling based formation of acetylenic carbon nanofiber thin films on various substrates and investigates their performance as active material in photoelectrochemical cells.

The manuscript is well-written, the data is of high quality. The multitechnique characterization provides strong support for the claims. The performance of such easy-to-produce devices is impressive and highlights convincingly the potential related to carbon materials incorporating diyne moieties. The remarkable improvement of the photocatalytic activity over other polymer-based

systems will impact the field.

The only remark I have is that I don't why the fabrication approach should be called "on-surface synthesis". The shown results do not give evidence that the polymerization takes place on the employed Cu surfaces. Also the formation of several hundred nm thick films is not consistent with the assumption that on-surface processes are essential for their formation. I suggest to simply remove the on-surface categorization. Also the Unit of time "s" should not be capitalized (see Figs. 4 & 5)

In summary, after these minor corrections, I full support the manuscript for publication in Nat Communs.

We address the concerns of Reviewer 1# as follows:

Comments:

Photo-induced water splitting is regarded as promising and attractive pathway for achieving solar-to-fuel conversion, and effective hydrogen evolution using earth-abundant and low-cost elements is with vital importance. In this manuscript, the authors have established an on-surface synthesis approach for the production of acetylenic carbon-rich nanofibers via the Cu-mediated Glaser polycondensation of 1,3,5-triethynylbenzene (TEB) on a variety of substrates for PEC hydrogen evolution. The result presented here is interesting. However, we still cannot recommend the publication of this manuscript in Nature Communication at this stage for the following reasons: 1) the production of H₂ has not been confirmed; 2) the mechanism of proton reduction is not provided; 3) the role of introduced DET unit is not very clear; the efficiency of this system is not astonishing; and 4) finally, the manuscript is finished in a hurry, lots of mistakes can be observed. The specific comments are below:

Response:

We greatly appreciate the reviewer for the valuable comments and suggestions to deepen the study in one specific area. Following the reviewer's suggestions, additional experiments have been performed and point-by-point responses to all the specific comments raised have been provided below. In the meantime, we would like to thank to reviewer to propose the four points, and we are able to address them in the revised version: 1) we observed H₂ production of PTEB as photocathode in PEC cell (2.53 μmol of H₂ evolution in 11 h reaction under AM 1.5G irradiation at 100 mW cm⁻²) as well as bulk PTEB powder in photocatalytic processes (*ca.* 10 μmol g⁻¹ in 3 h reaction under 300 W Xenon lamp, λ > 420 nm); 2) the mechanism for proton reduction was added with experiments as well as discussions. In short, we observed a significant enhancement of photocurrent, by adding of electron scavengers Cu²⁺ (it reacts with electron to yield Cu⁺, *Nat. Mater.* 2015, 14, 505). This is due to the reduction of Cu²⁺ that is thermodynamically and kinetically more facile than the reduction of water, which confirms the proton reduction property of PTEB; 3) we have demonstrated that by introducing PDET segment in the PTEB structure, the light absorption range was obviously enlarged. In addition, the electronic band structure and charge transfer capability for water reduction were significantly improved on the PTEB_{1.3}-*co*-PDET₁ photocathode. We agree with the reviewer's comments that the photocurrent efficiency is not astonishing when comparing to inorganic materials. Nevertheless, the achieved PTEB is superior to the-state-of-art metal-free/polymeric materials, such as *g*-C₃N₄ that normally delivers photocurrent density in the range of 0.1-1 μA cm⁻². Furthermore, a brief comparison to the recently reported metal-free photocathode materials is presented in Table S1 in the Supplementary Information. On the other hand, in contrast to inorganic semiconductors (*e.g.*, TiO₂,

ZnO, Fe₂O₃ and WO₃) which have been intensively investigated as photoelectrodes for PEC water splitting, organic (polymer) semiconductors based on earth-abundant elements, featuring with tunable energy levels, low manufacturing cost, abundance, and environmental sustainability (*Acc. Chem. Res.* 2010, 43, 1063) remain under development. In this regard, this work highlights the promise of utilizing acetylenic carbon-rich materials as novel organic photocathode material for water reduction. 4) We are sorry for the mistakes; all of them have been corrected following the reviewer's suggestions in the revised manuscript.

Question 1:

According to the reported results (*J. Phys. Chem. C* 2012, 116, 9398–9404), the potentials vs Ag/AgCl can be converted to the reversible hydrogen electrode (RHE). In order to convert a bias vs. Ag/AgCl into vs. RHE, the temperature and pH value of the solution should be provided.

Response:

We thank this reviewer for the constructive comment. We have converted the potentials vs. Ag/AgCl to vs. RHE. And the temperature (*ca.* 25 °C) and pH value (pH = 6.8) of the PEC cell have been provided in the experimental section in the revised manuscript.

Question 2:

In Fig. 4c, PTEB nanofiber based photocathode can give a photocurrent of ~10 μA cm⁻² at a bias of -0.3 V vs. Ag/AgCl. However, the photocurrent density still keeps at ~10 μA cm⁻² when the external bias increased to -0.6 V vs. Ag/AgCl (Fig. 5c), why? For a better comparison, the photocurrent responses at -0.4, -0.5 and -0.6 V vs. Ag/AgCl should be added in Fig. 4d.

Response:

Following the reviewer's suggestion, new data were measured and shown in Fig. R1. It can be seen that there is no significant change on the photocurrent density when the voltage was applied from 0.3 V to 0 V vs. RHE (*i.e.* -0.3 – -0.6 V vs. Ag/AgCl), indicating that the PTEB nanofiber photocathode reached its saturation photocurrent below 0.3 V vs. RHE (*i.e.* -0.3 V vs. Ag/AgCl). A similar phenomenon was observed in WSe₂ thin films electrodes in PEC hydrogen production (*Nat. Commun.* 2015, 6, 7596), as well as other metal-free semiconductors, such as red phosphorus (*Angew. Chem. Int. Ed.* 2016, 55, 9580). The new results have been included in Fig. 4d in the revised manuscript.

Figure R1 | Photocurrent densities vs. time for PTEB with different applied bias potentials (from 0.5 V to 0 V vs. RHE) under intermittent irradiation.

Question 3:

The authors claim that electrons accumulated in PTEB are consumed to form H₂ by reducing water. So, my question is, have you directly detected H₂ gas using gas chromatography? If yes, please give us the corresponding IPCE value of this system? Also, if H₂ gas does evolve here, could you explain us the active sites or mechanism of proton reduction on the metal-free material?

Response:

We thank the reviewer for the constructive comments. In order to detect the H₂ gas in the PEC cell, a PTEB nanofibers film (*ca.* 1 cm²) of *ca.* 230 nm was prepared on Ti foil as photocathode. As shown in Fig. R2a, moderate amount of H₂ evolution (2.53 µmol) could be measured in 11 h reaction by using gas chromatography at the PTEB photocathode with 0 V vs. RHE (*i.e.* -0.6 V vs. Ag/AgCl) of applied bias in 0.01 M Na₂SO₄ solution under AM 1.5G irradiation (100 mW cm⁻²). Furthermore, a maximum IPCE value of PTEB photocathode was measured to be 1.06% at 460 nm (Fig. R2b). Such value is well comparable to those in previous reports of polymeric carbon nitrides (*J. Mater. Chem. A* 2015, 3, 3281; *J. Am. Chem. Soc.* 2009, 131, 1680), red phosphorus (*Angew. Chem. Int. Ed.* 2016, 128, 9732), and even some transition metal dichalcogenides (*e.g.* WSe₂, *Nat. Commun.* 2015, 6, 7596) as well as p-type metal-containing counterparts (*e.g.* Cu-Ti-O: *Nano Lett.* 2008, 8, 1906). Although the achieved value is still lower than those of state-of-the-art inorganic hybrid photocathodes (*e.g.* 40% for sophisticated Cu₂O/ZnO:Al/TiO₂/Pt, *Nat. Mater.* 2011, 10, 456), our result suggests that there is considerable scope for the further improvement in either photoelectrode layout, co-catalyst, or both.

Figure R2 | (a) Amount of evolved H₂ (black spots) and recorded charge carrier (red line) on PTEB electrode in PEC cell with 0 V vs. RHE of applied bias in 0.01 M Na₂SO₄ under AM 1.5G irradiation (100 mW cm⁻²). The experimental and theoretical values represent the observed and expected amount of hydrogen evolved, assuming a quantitative Faradaic yield for H₂ formation. **(b)** The incident-photon-to-current-conversion-efficiency (IPCE) spectrum of the PTEB photocathode measured at 0 V vs. RHE and in 0.01 M Na₂SO₄ under AM 1.5G irradiation. A maximum IPCE value of 1.06% was achieved at 460 nm on PTEB photocathode.

These new results have been included in the Supplementary Figures 15 and 22.

Regarding to the active sites and mechanism of proton reduction, the energy band structure of PTEB nanofibers was resolved by ultraviolet photoelectron spectroscopy (UPS). The results indicate that the electronic band structure of PTEB nanofiber (*i.e.* -0.68 vs. NHE, Fig. 3d) is higher than the theoretical potential of hydrogen evolution (0 V vs. NHE), indicating that the PTEB photocathode is energetically and thermodynamically favourable for water reduction. It has been reported that the carbon-rich polymer poly(p-phenylene) (PPP) was able to catalyse the one-electron reduction of aryl aldehydes to generate a ketyl intermediate under visible light irradiation (*Eur. J. Org. Chem.* 2012, 6187). Recently, the acetylenic carbon-rich poly(diphenylbutadiyne) (PDPB) nanofiber was reported to show high photocatalytic activity (degradation of pollutants) under visible light without the assistance of sacrificial reagents or precious metal co-catalysts (*Nat. Mater.* 2015, 14, 505). And the generation of photoexcited electron (to generate reactive oxygen species O₂^{•-}) on the PDPB has been well demonstrated by control experiments with various scavengers (*e.g.*, Cu²⁺ and isopropanol). In addition, Cooper *et al.* have proved that carbon-rich polymers (contains carbon and hydrogen only) can be used, without the addition of additional metal cocatalysts, for the photocatalytic H₂ evolution from water splitting (*J. Am. Chem. Soc.* 2015, 137, 3265; *Nature* 2015, 521, 41). All of these works give solid demonstration that (acetylenic) carbon-rich polymers are a promising class of organic semiconductors for photocatalytic (one-electron) proton reduction. Nevertheless, it remains challenging thus far to gain fundamental understanding the exact active sites in carbon-rich materials for photocatalytic H₂ production.

Question 4:

In order to directly confirm the proton reduction property of PTEB, we think that photocatalytic hydrogen evolution performance of this material can be evaluated in the presence of sacrificial reagents, such as ascorbic acid, methanol, or triethylamine etc.

Response:

Indeed, we agree with the reviewer that the proton reduction property of PTEB should be proved. Considering that it is almost impossible to collect enough quantity (*e.g.* 50 mg for single measurement) of PTEB nanofibers powders via detaching from solid substrate to measure photocatalytic performance for H₂ evolution, we evaluated the effects of the sacrificial reagents on photocurrent density of PTEB photocathode in PEC cell. In this respect, the transient photocurrent density of PTEB photocathode was examined in a 0.01 M Na₂SO₄ electrolyte (pH 6.8) and an electron scavenger (10⁻⁵ M Cu²⁺, it reacts with electron to yield Cu⁺, *J. Fluoresc.* 2009, 19, 723; *Nat. Mater.* 2015, 14, 505). The reduction of Cu²⁺ is thermodynamically and kinetically more facile than the reduction of water. As such, the cathodic photocurrent density at PTEB nanofibers is noticeably accelerated (to *ca.* 18 μA cm⁻² at 0.3 V *vs.* RHE) in the presence of Cu²⁺ (Fig. R3). The significant enhancement of photocurrent indicates that the competitive reaction of excess photogenerated electron with Cu²⁺ decreases the recombination rate, leading to more available excess hole at photoelectrode. In addition, the photocurrent density increased by adding 2% diethylamine (to *ca.* 11 μA cm⁻² at 0.3 V *vs.* RHE) or ascorbic acid (to *ca.* 13 μA cm⁻² at 0.3 V *vs.* RHE), known as hole scavengers, which enhanced the consumption of holes and increased the photoelectrons transfer to cathode. These results support that the PTEB nanofibers have the proton reduction capability.

The new results have been included in Supplementary Figure 21.

Figure R3 | Transient photocurrent density vs. time of PTEB photocathode at a bias of 0.3 V vs. RHE (*i.e.* -0.3 V vs. Ag/AgCl) under intermittent irradiation in various electrolytes: 0.01 M Na₂SO₄ (black curve), hole scavenger diethylamine (2%) in 0.01 M Na₂SO₄ (red curve), hole scavenger ascorbic acid (2%) in 0.01 M Na₂SO₄ (blue curve), and electron scavenger Cu²⁺ (10⁻⁵ M) in 0.01 M Na₂SO₄ (green curve).

In addition, we also tested the photocatalytic hydrogen evolution experiments of bulk PTEB directly synthesized in solution by ever reported method (*Adv. Mater.* 2014, 26, 8053). Although the structure of PTEB was confirmed by Raman results, which is identical to the PTEB nanofibers synthesized on the surface, the morphology of bulk PTEB from solution approach features with microscopic-porous particles rather than nanofibers. In addition, the bulk PTEB powder exhibited very poor photocatalytic performance (*ca.* 10 μmol g⁻¹ in 3 h reaction, 300 W Xenon lamp, λ > 420 nm) for H₂ evolution in the presence of triethylamine (TEA) or triethanolamine (TEOA) as sacrificial reagents. The discrepancy in the performance of PTEB in PEC and photocatalytic cell is probably due to the larger particle size of bulk PTEB favouring higher recombination rate of photoelectrons and holes. The strong dependence of the photocatalytic activity on the size and morphology has also been observed in other semiconductors such as poly(diphenylbutadiyne) (*Nat. Mater.* 2015, 14, 505), TiO₂ (*Chem. Lett.* 2009, 38, 238), BiVO₄ (*ACS Appl. Mater. Interfaces* 2017, 9, 505). etc., and is still a matter of further investigation.

A broad application of PTEB related materials in solar energy conversion remain to be investigated in the future. The current work has been focused on the development of novel interfacial synthesis of acetylenic polymer nanofibers through Cu-surface mediated Glaser coupling reaction. It is straightforward to test the performance of as-prepared PTEB nanofibers on conductive substrates as photoelectrodes in PEC cell. In principle, the photocatalytic cell is a different reaction, which needs special in-solution synthetic approach, and the choice of co-catalyst and sacrificial agents is rather

critical. Our work has demonstrated that bulk PTEB powder could be used in photocatalytic cell for H₂ evolution, whereas the morphology and performance still need further optimization.

Question 5:

The resolution/magnification of TEM image in Fig. 1d is too low to give the corresponding size distribution of PTEB in Fig. 1e. We do think that TEM images with a much higher resolution are necessary.

Response:

Following the reviewer's suggestion, in the revised manuscript we have replaced Fig. 1d with a new TEM image (Fig. R4) of higher resolution.

Figure R4 | Transmission electron microscopy (TEM) image of PTEB grown on a Cu grid, scale bar: 200 nm.

Question 6:

In Fig. 1f, the author assigned the peaks at 989 cm⁻¹ and 1581 cm⁻¹ respectively to the ring breathing and ring stretching modes of the aromatic moieties. We think that proper references should be cited here to support your claim.

Response:

We thank the reviewer for pointing this out. We have added *Sensors* 2011, 11, 11510 to reference 38 in the revised manuscript. In addition, more clear demonstrations from DFT calculation to support the assignment have been updated in Supplementary Figure 9.

Question 7:

It can be observed that the morphology of the obtained PTEB on different substrates is different. So, what's the reason? And could you control the morphology of PTEB through a facile pathway? Besides, have you explored the influence of morphology on PEC performance?

Response:

We thank the reviewer for the constructive comments. As we have shortly described in page 7 text "*Based on the above results, we propose a mechanism according to a previous model describing the synthesis of polyaniline nanofibers on a solid substrate via stepwise electrochemical polymerization. These active sites minimize the interfacial energy barrier for the subsequent growth of PTEB nanofibers on the solid substrates*". Therefore, the reasons for the varied PTEB morphologies on different substrates could be put forward as: different substrates have different surface energy and roughness that resulted in different nucleation rates, which not only affect the reaction kinetics of Glaser coupling, but also the approximation of monomer to the surface of substrate. Certain morphology of PTEB is controllable, and can be readily reproduced on each substrate.

We further measured the PEC performance of PTEB photocathodes derived from various substrates, such as Cu foil, graphite, fluorine doped tin oxide (FTO) and Ni foil. Compared to the PTEB grown on Ti foil, the photocathode from the PTEB grown on graphite provided much lower photocurrent density ($3.9 \mu\text{A cm}^{-2}$ at 0.3 V vs. RHE), whereas that of on Cu foil showed the highest value with *ca.* $18.6 \mu\text{A cm}^{-2}$ at 0.3 V vs. RHE (*i.e.* $-0.3 \text{ V vs. Ag/AgCl}$). This is reasonable when the morphologies of the PTEB nanofibers were concerned (Fig. 2 and Fig. S11), since the graphite surface only yielded large bundles (which are not favourable for charge transfer) of PTEB fibres with diameter of *ca.* 40 nm (Fig. 2b), in contrast to that obtained on Cu surface (*ca.* 10 nm, Fig. 1e). The strong effect of size and morphology of the structure on the photocatalytic activity of semiconductors has been extensively investigated, such as poly(diphenylbutadiyne) (*Nat. Mater.* 2015, 14, 505), TiO_2 (*Chem. Lett.* 2009, 38, 238), BiVO_4 (*ACS Appl. Mater. Interfaces* 2017, 9, 505), etc. Unfortunately, the Cu substrate may not be suitable as electrode for PEC reaction, due to the possibility of electrochemical corrosion of Cu in the cell.

The new results have been included in Supplementary Figure 22.

Figure R5 | Transient photocurrent density vs. time of PTEB photocathode prepared on various conductive substrates (Cu foil, graphite, FTO glass, and Ni foil) at a bias of 0.3 V vs. RHE (*i.e.* -0.3 V vs. Ag/AgCl) under intermittent irradiation in 0.01 M Na₂SO₄.

Question 8:

In page 9, the authors claim that “we note that there is no obvious difference in the absorption spectra of the PTEB nanofibers grown on different substrates.” Please provide us the UV-vis absorption spectra of PTEB grown on different substrates.

Response:

Following the suggestion from the reviewer, the UV-vis absorption spectra of PTEB grown on quartz, glass, fused Si and Cu were measured as shown in Fig. R6. All of the spectra exhibit identical peaks and absorption edge.

The new results have been included in Supplementary Figure 14.

Figure R6 | UV-vis absorption spectra of PTEB films grown on various substrates: fused Si (black curve), glass (red curve), FTO glass (green curve) and quartz (blue curve).

Question 8:

Do the authors think that the thickness of PTEB would exert an influence on PEC performance? You'd better provide us direct evidences.

Response:

We thank the reviewer for the constructive comment. Following the suggestion, we have measured the transient photocurrent density vs. time of PTEB photocathode with various thickness (*i.e.*, 95 nm, 230 nm, 390 nm, and 560 nm) by controlling the reaction time, as shown in Figure R7. In short, a photocurrent density of $4.2 \mu\text{A cm}^{-2}$ could be obtained on a 95 nm thick PTEB layer at a bias of 0.3 V vs. RHE. The value reached *ca.* $10 \mu\text{A cm}^{-2}$ when the layer of PTEB nanofibers film increased to *ca.* 230 nm thick, since the light absorption was improved. However, the photocurrent dropped continuously when the thickness of PTEB was further increased to 390 nm and 560 nm, respectively. This result can be attributed to the long-distance electron transfer from PTEB surface to electrode. Therefore, the optimization of PTEB film thickness to reach equilibrium between the light adsorption capability and charge transfer efficiency is necessary to achieve the optimized PEC performance.

The new results have been included in Supplementary Figure 16.

Figure R7 | Transient photocurrent density vs. time of PTEB photocathode of various thickness (95, 230, 390, and 560 nm) at a bias of 0.3 V vs. RHE (i.e. -0.3 V vs. Ag/AgCl) under intermittent irradiation in 0.01 M Na₂SO₄.

Question 10:

The PEC performance of PTEB can be further improved by introducing DET unit into the backbone. We think that the author must explain us the intrinsic reasons of the enhanced PEC performance. Is it merely a matter of enhanced light absorption? Will the band gap, CB and VB, change either? We also think that the conductivity of PTEB before and after DET introduction should be compared.

Response:

We thank the reviewer for the constructive comment. The pristine PTEB nanofibers could only absorb light below 500 nm, thus the original motif for introducing DET section was to improve the light absorption. We are sorry that we have missed the investigation in other factors that may also contribute to the PEC performance of PTEB_{1.3-co}-PDET₁ photocathode. Following the reviewer's suggestion, we measured the CB, VB, and conductivity of the PTEB_{1.3-co}-PDET₁, and the data are shown in Fig. R8.

We found that the PTEB_{1.3-co}-PDET₁ has a lower conduction band (CB) of -1.47 V vs. NHE than that of pristine PTEB (-0.68 V) as well as than the theoretical potential of water reduction (0 V vs. NHE). Also, the CB of PTEB_{1.3-co}-PDET₁ is much higher than previously reported *g*-C₃N₄ (-1.12 V, *Nat. Mater.* 2009, 8, 76), Hittorf red phosphorus (-0.25 V, *Appl. Catal. B* 2012, 111-112, 409), and fibrous red phosphorus (-0.9 V, *Angew. Chem. Int. Ed.* 2016, 55, 9580). This result suggests that the introduction of DET in the PTEB network is more energetically favourable for water reduction (*Nat.*

Commun. 2014, 5, 3605). The larger energy gap between the CB edge and the H^+/H_2 potential for $PTEB_{1.3}\text{-}co\text{-}PDET_1$ is beneficial for water reduction, which should be one of the reasons for enhanced photocatalytic activity of $PTEB\text{-}co\text{-}PDET$ (Fig. 5c).

Figure R8 | Characterization of the optical and electronic structure of $PTEB_{1.3}\text{-}co\text{-}PDET_1$ copolymer film. (a) UPS spectra (black curve). The dashed red lines mark the baseline and the tangents of the curve. The intersections of the tangents with the baseline give the edges of the UPS spectrum from which the UPS width is determined. (b) UV-vis absorption spectra. Inset: digital photograph of the $PTEB_{1.3}\text{-}co\text{-}PDET_1$ film (ca. 230 nm) transferred to a PET substrate. (c) $(h\nu\alpha)^2$ vs. $h\nu$ curve (black curve). The horizontal dashed red line marks the baseline; the other dashed line marks the tangent of the curve. The value at the intersection is the bandgap: $E_{bg} = 1.93$ eV.

To investigate the conductivity of PTEB before and after DET introduction, two-electrode conductivity measurements over $2.5\ \mu\text{m}$ channels were performed on PTEB and $PTEB_{1.3}\text{-}co\text{-}PDET_1$ films, respectively, deposited on commercial organic field-effect transistor (OFET) substrates (Fraunhofer IPMS, Dresden). As shown in Fig. R9, I - V curves for both the PTEB and $PTEB_{1.3}\text{-}co\text{-}PDET_1$ films exhibit semiconductor-like characteristics (*Macromolecules* 2008, 41, 7383; *J. Am. Chem. Soc.* 2017, 139, 11666). R of the samples was estimated from the inverse slope of the I - V curve. When the I - V curve is not linear, the slope of the curve was estimated from the linear fit of the curve. Thus, the conductivity, σ , of the $PTEB_{1.3}\text{-}co\text{-}PDET_1$ copolymer film was found to be of 1.9×10^{-5} S/cm (see Fig. R9b), which is about an order of magnitude greater than that of pristine PTEB film (*i.e.* 3.0×10^{-6} S/cm; Fig. R9b).

Given the above results, it is reasonable to conclude that by introduction of PDET segment in the PTEB structure, not only the light absorption range was enlarged, but also the electronic band structure and charge transfer capability for water reduction were improved for the PTEB_{1.3}-co-PDET₁ photocathode.

The new results have been included in Supplementary Figures 19 and 20.

Figure R9 | Conductivity comparison of the PTEB and PTEB_{1.3}-co-PDET₁ films. Representative *I*-*V* characteristic curves of (a) PTEB and (b) PTEB_{1.3}-co-PDET₁ on the 2.5 μm channel OFET device. In both panels, the average thickness of the polymer films is *ca.* 230 nm, and both films exhibit semiconductor-like characteristics.

Question 11:

11. In Supplementary Figure 2a, the unit of x-axis should be hour (h). In Supplementary Figure 2b, we think that the unit of x-axis should be micron (μm).

Response:

We are sorry for the mistake. The unit has been corrected following the suggestion.

Question 12:

12. In Supplementary Figure 3, a cross-section SEM image of PTEB on a copper coated Si wafer was provided. We think that, for a better view of the structure, corresponding elemental mapping should be provided.

Response:

Following the suggestion, elemental mapping images have been measured at the cross-section of PTEB grown on Cu coated Si wafer. As shown in Fig. R10, the energy dispersive X-ray (EDX) mapping suggests a clear contrast of different layers (PTEB, Cu and Si) on the sample cross-section.

The new results have been included in Supplementary Figure 3b-g.

Figure R10 | Cross-section SEM image and energy dispersive X-ray (EDX) elemental mapping images of PTEB on a copper coated Si wafer. A PTEB layer of *ca.* 230 nm thick can be identified in the (a) Cross-section SEM image of the sample. Further EDX maps show clear contrast of different layers on the sample: (b) full elements map, (c) carbon, (d) copper, (e) silicon and (f) oxygen. (g) The corresponding EDX spectrum of (b) measured at 3 kV acceleration voltage.

Question 12:

This manuscript is finished in a hurry. As a result, there are many style mistakes. For example, the unit of time (second) should be written in s, not S (Figure 4); and the first letter should be capitalized of each sentence (see corresponding mistakes in the caption of every figure).

Response:

We are sorry for the caused confusion and mistakes. The unit in Fig. 4 has been corrected in the revised manuscript, and all the other mistakes in figure caption have been corrected as well.

We address the concerns of Reviewer 2# as follows:**Comments:**

The paper describes the preparation, characterization and photoelectrochemical response of a polybutadiene polymer deposited onto metallic and carbon substrates. The material is prepared by a Cu-catalyzed reaction (Glaser) between terminal aryl acetylenes. The key finding reported is that the modified electrodes exhibit a cathodic photocurrent arising from reduction of protons (however production of H₂ is not confirmed experimentally). The paper is reasonably well-written, and the work is novel. While the paper does contain somewhat exaggerated claims (see below), the work should be of sufficient interest to warrant publication in the journal. I recommend publication pending revision to address a number of significant comments as outlined below.

Response:

We greatly appreciate the valuable comments from the reviewer. All the suggestions from the reviewer have been carefully addressed and changes have been made accordingly.

Question 1a:

Exaggerated/unsubstantiated claims:

A photocurrent of 10 μ A under AM1.5 illumination corresponds to <0.1% overall quantum efficiency. Given this, I recommend that the authors remove all hyperbole such as “highly active” (title), “excellent photocatalytic activity” (abstract), “excellent charge transport” (unsubstantiated by any conductivity or mobility measurements), “excellent polymeric photocathode”, etc.

Response:

These words were used since we found that the PEC performance of PTEB photocathode is superior to that of $g\text{-C}_3\text{N}_4$ measured under the same conditions (Fig. 4c), and other reported metal-free photocathode materials as listed in Table 1 in Supplementary Information. However, we agree with the reviewer that such performance is not exciting if the overall quantum efficiency was calculated, which is much less than inorganic semiconductors. Following the suggestion from the reviewer, we have removed or modified these words in the revised version.

Question 1b:

Water-splitting has not been demonstrated. At best, proton-reduction is accomplished, but even this is not proved. I strongly recommend that evidence for H_2 production is provided before the paper can be accepted. Even if H_2 production is demonstrated, I recommend that the use of “water splitting” is not used in the paper to describe what amounts to a photocathodic reaction. Water splitting implies that the material is able to carry out the reaction, $\text{H}_2\text{O} \rightarrow \text{H}_2 + \text{O}_2$, and while O_2 may be generated at a counter electrode, this is under the (unknown) applied bias of a 3-electrode configuration using a potentiostat, and the actual potential applied to the auxiliary electrode is unknown.

Response:

We thank the reviewer for the constructive comment. All the words “water splitting” in describing the performance of PTEB have been replaced by “water reduction” and “hydrogen production” in due course in the revised manuscript.

In order to detect the H_2 gas in the PEC cell, a PTEB nanofibers film (*ca.* 1 cm^2) of *ca.* 230 nm was prepared on Ti plate as photocathode. As shown in Fig. R11, moderate amount of H_2 evolution (2.53 μmol) could be measured in 11 h reaction by using gas chromatography with thermal conductivity detector (TCD, N_2 carrier) at the PTEB photocathode with 0 V *vs.* RHE (*i.e.* -0.6 V *vs.* Ag/AgCl) of applied bias in 0.01 M Na_2SO_4 solution under AM 1.5G irradiation (100 mW cm^{-2}).

The new results have been included in Supplementary Figure 22.

Figure R11 | Amount of evolved H₂ (black spots) and recorded charge carrier (red line) on PTEB electrode in PEC cell with 0 V vs. RHE of applied bias in 0.01 M Na₂SO₄ under AM 1.5G irradiation (100 mW cm⁻²). The experimental and theoretical values represent the observed and expected amount of hydrogen evolved, assuming a quantitative Faradaic yield for H₂ formation.

In addition, we also conducted the photocatalytic hydrogen evolution experiments of bulk PTEB, which refer to the Response to Question 4 of reviewer #1.

Question 2:

2) Absolute monochromatic illumination quantum yield and IPCE spectrum should be reported. This will highlight the true potential for this and related materials to be useful in solar energy conversion.

Response:

We thank the reviewer for the constructive comments. The IPCE spectrum was measured on the PTEB photocathode in PEC cell at a potential of 0 V vs. RHE (*i.e.* -0.6 V vs. Ag/AgCl) in 0.01 M Na₂SO₄ solution using the Xenon lamp with specific wavelength filters to select the required wavelength of light (AM 1.5G). A maximum IPCE value of 1.06% at 460 nm was obtained on PTEB photocathode (Fig. R12), which is comparable to those in existing reports of polymeric carbon nitrides (*J. Mater. Chem. A*, 2015, 3, 3281; *J. Am. Chem. Soc.* 2007, 131, 1680) and even some transition metal dichalcogenides (*e.g.* WSe₂, *Nat. Commun.* 2015, 6, 7596) and p-type metal-containing counterparts (*e.g.* Cu-Ti-O: *Nano Lett.* 2008, 8, 1906).

The new results have been included in Supplementary Figure 15.

Figure R12 | The incident-photon-to-current-conversion-efficiency (IPCE) spectrum of the PTEB photocathode measured at 0 V vs. RHE and in 0.01 M Na₂SO₄ under AM 1.5G irradiation. A maximum IPCE value of 1.06% was achieved at 460 nm on PTEB photocathode.

However, if we understood the reviewer's comment correctly, the absolute monochromatic illumination quantum yield should be measured in a photocatalytic cell. This is particularly challenging for the present work, since a relatively large amount (*ca.* 50 mg for a single measurement) of material is required for the photocatalytic H₂ evolution measurement, and it is nearly impossible to prepare such amount of PTEB nanofibers via the current on-surface synthesis approach. Instead, we measured the photocatalytic H₂ evolution performance of bulk PTEB that was directly synthesized in solution. Although H₂ evolution was observed from the bulk PTEB powders (refer to the Response to Question 4 of reviewer #1), the yield is too low to measure trustable data of monochromatic illumination quantum yield.

Question 3:

The pH of the electrolyte solution is not reported, and this is quite an important parameter.

Response:

We have added the pH value (pH= 6.8) of the electrolyte solution in the experimental part.

Question 4:

Throughout the paper, the authors imply that the polybutadiene has a "high conjugation length" (text, p. 6). This is not at all likely. The backbone structure consists of phenyl-CC-CC-phenyl units linked through meta-connections. This type of structure is not very strongly delocalized (see for example, DOI: 10.1021/ja029489h). Singlet excitons in such meta-linked phenyl-ethylene structures are confined to very short segments, and if there is much delocalization between multiple chains

(interchain excitons), these also will be spatially confined. The same can be said for the charged states of the material (polarons). Thus, transport in the material will be relatively poor, because the dominant mechanism is hopping. Therefore, the authors should restrict their use of exaggerated claims about “excellent transport”, etc. Indeed, the EIS plot in Figure 4e suggests that the material is resistive (*e.g.*, poor transport).

Response:

We appreciate the valuable comments and literature from the reviewer. We agree with the reviewer that the conjugation property of PTEB network is not that good. Thus, the words “high conjugation length” has been replaced by “high polymer chain length” in page 5 (text) in the revised manuscript.

We note from the literature (*J. Am. Chem. Soc.* 2003, 125, 9288, reviewer suggested) that phenylacetylene dendrimers are weakly coupled in their equilibrium ground-state due to the meta-conjugation feature of phenylacetylene unit. Moreover, in that work, the authors also stated that phenylacetylene could become strongly coupled (enhanced charge transfer) in the excited state. For example, in the Fig. 2a of the reference paper, the calculation results demonstrate that the excited-state geometry (*e.g.* 3-H) leads to a cumulenic structure in phenylacetylene, which would make a better conductor. We agree with the reviewer that indeed the charge transport inside the PTEB structure is not excellent, as evidenced by the EIS plots (Fig. 4e). The conductivity of pristine PTEB nanofibers film was found to be 3.0×10^{-6} S/cm as a typical semiconductor (refer to the Response to Question 10 of reviewer #1), which is close to the pristine (undoped) P3HT film (*ca.* 10^{-5} S/cm; *Macromolecules* 2008, 41, 7383). Therefore, the words “excellent transport” has been replaced by “enhanced transport” in page 14 (text), and similar claims have been changed in the revised manuscript.

Question 5:

The Glaser reaction is catalyzed by Cu(I), and the authors are using metallic Cu (copper(0)). Presumably dioxygen is the oxidant? The authors should make this clear.

Response:

We are sorry for the caused confusion. We agree with the reviewer that the classical Glaser coupling is catalysed by Cu(I) and an additional oxidant (*e.g.* O₂). Lately, the Cu(II) salts can also be used as catalysts for Glaser coupling reaction (*Tetrahedron Letters*, 2011, 52, 3485). With respect to reference 32 (*Polym. Chem.* 2015, 6, 2726) in the manuscript, catalytic Cu(I/II) species can be continuously generated from metallic Cu surface in a polar liquids or alkaline solutions. These Cu(I/II) species could catalyse Glaser coupling at the interface between Cu wafer (foil) and facing substrates.

Therefore, we have modified the description in the main text with: “*Since both Cu^I and Cu^{II} salts have been widely used as catalysts for Glaser coupling reaction (Tetrahedron Letters, 2011, 52, 3485, Angew. Chem. Int. Edit. 2000, 39, 2633), we expected that the Cu species generated from the metallic copper surface would be able to catalyze the Glaser C-C coupling reaction*”

Question 6:

p. 10, value used to convert from vacuum level scale to NHE (-4.44 V) is not correct. This means the redox potentials shown for the material in Figure 3d are not correct. The oxidation and reduction potentials should be shifted negative by at least 400 mV, see: 10.1002/adma.201004554. This does not change the major conclusion that the polymer has the thermodynamic potential to split water, but the values reported in the diagram are incorrect.

Response:

We agree with the reviewer that oxidation and reduction potentials should be negatively shifted. In the revised manuscript, we have negatively shifted the potentials by 400 mV following the suggestion and the literature (*Adv. Mater.* 2011, 23, 2367)

Question 7a:

Minor comments:

Labels a, b, c in the figures are too small and are difficult to discern, especially in Figure 1. All instances should be enlarged.

Response:

Following the suggestion from the reviewer, these labels in figures have been enlarged.

Question 7a:

Binding is misspelled in x-axis in Figure 1 (2 instances)

Response:

The mistakes have been corrected in Fig. 1g and h.

Question 7c:

Planar is misspelled on p. 6 (planer)

Response:

The error has been corrected in the revised manuscript.

Question 7d:

What is the value alpha (absorption coefficient) for the material (equation on p. 10)?

Response:

The absorption coefficient (α) can be calculated by $\alpha = (1/t) \ln [(1-R)^2 / T] = 2.303 A / h$ (*Superlattice. Microst.* 2016, 89, 153), where T and R are the transmission and reflection, A is absorbance and h is the thickness of PTEB film. In the case of PTEB (*ca.* 230 nm) at absorption peak 460 nm ($A = 0.34$) wavelength, we can derive the value $\alpha = 0.0034 \text{ nm}^{-1}$.

Question 7e:

Text should make it clear that the photoelectrochemistry is carried out in a 3-electrode cell with applied bias relative to the reference electrode (not the counter/auxilliary electrode).

Response:

We have added the demonstration “*The PEC characterization was carried out in a three-electrode cell with applied bias relative to the reference electrode (Ag/AgCl)*” in the text in page 11 in the revised manuscript.

Question 7f:

p. 11, strike “Remarkably”; as noted above 10 μA is hardly remarkable.

Response:

Following the suggestion from the reviewer, the word “Remarkably” has been removed.

Question 7e:

Figure 5: molecular structure for the copolymer is misleading. It is not known whether this is a truly alternating co-polymer structure. The relative loading of the co-monomer is not determined. The structure should show the subunits within parenthesis () with x and y subscripts.

Response:

The reviewer is correct in pointing out that the co-polymer PTEB-*co*-PDET is not really an alternating structure between TEB and DET, since we did not apply any special strategy to precisely control each unit during the polymerization process. The PTEB-*co*-PDET was obtained by normal random copolymerization of TEB (0.017 mmol) and DET (0.013 mmol). Finally, the relative loading of each monomer in the PTEB-*co*-PDET structure was determined by EDX spectrum (Fig. R13), which gave a clearer structure with subunits PTEB_{1.3}-*co*-PDET₁. Following the suggestion from reviewer, suitable subscripts have been added to the structure in Fig. 5.

The new results have been included in Supplementary Figure 18.

Figure R13 | EDX spectrum and morphology of PTEB_{1.3}-*co*-PDET₁ copolymer film. (a) EDX spectrum of PTEB_{1.3}-*co*-PDET₁ on SiO₂/Si wafer measured at 3 kV acceleration voltage. The ratio of PTEB and PDET in the copolymer structure can be estimated by the ratio of carbon and sulphur from the EDX spectrum. Therefore, the copolymer can be more precisely defined as PTEB_{1.3}-*co*-PDET₁. (b) SEM image reveals that the PTEB_{1.3}-*co*-PDET₁ copolymer shows larger nanofiber (bundle) morphology than PTEB.

We address the concerns of Reviewer 3# as follows:

Reviewer #3 (Remarks to the Author):

Comments:

The manuscript by Tao Zhang et al reports the Cu-mediated Glaser-coupling based formation of acetylenic carbon nanofiber thin films on various substrates and investigates their performance as active material in photoelectrochemical cells.

The manuscript is well-written, the data is of high quality. The multitechnique characterization provides strong support for the claims. The performance of such easy-to-produce devices is impressive and highlights convincingly the potential related to carbon materials incorporating diyne moieties. The remarkable improvement of the photocatalytic activity over other polymer-based systems will impact the field.

The only remark I have is that I don't why the fabrication approach should be called "on-surface synthesis". The shown results do not give evidence that the polymerization takes place on the employed Cu surfaces. Also the formation of several hundred nm thick films is not consistent with the assumption that on-surface processes are essential for their formation. I suggest to simply remove the on-surface categorization. Also the Unit of time "s" should not be capitalized (see Figs. 4 & 5).

In summary, after these minor corrections, I full support the manuscript for publication in Nat Communs.

Response:

We greatly appreciate the reviewer for the positive and constructive comments from the reviewer. Following the suggestions, we have changed the word "on-surface" to "Cu-surface mediated" in the revised version. And the mistakes in Figs. 4 & 5 have been corrected.

Reviewers' comments:

Reviewer #1 (Remarks to the Author):

First of all, I would like to thank the authors' efforts to make this manuscript much better and most of the proposed questions have been clarified. However, following concerns are still not clear:

1. As H₂ gas, albeit very few, has been observed in the revised manuscript, the corresponding faradic efficiency of PEC H₂ evolution should be provided.
2. Although the authors have enumerated multiple examples of employing carbon-rich polymers for photo-induced reduction reactions, such as aryl aldehyde reduction, Cu(II) reduction and pollutant degradation etc., the reduction of proton to hydrogen gas is a totally different reaction from the above-mentioned examples. As known to all, the reduction of proton to molecular hydrogen gas in general goes through the process of proton adsorption-reduction-hydrogen adsorption. Hence, the discussion of active sites for proton reduction by using this polymer is still not clear in my opinion. The authors must clearly clarify the true active sites for proton reduction.
3. In fact, bulk PTEB demonstrates negligible photocatalytic hydrogen evolution in the presence of sacrificial reagents. This is a very strange phenomenon! As shown in Figure 3d, PTEB acquires the ability for both proton reduction and even water oxidation. It can't reduce proton to hydrogen in the presence of sacrificial reagent, which is much easier to be oxidized than water. Why? Also, this result questions the true active sites of proton reduction in the PEC system. Two possibilities can be proposed: (1) do you think that residual Cu species in the polymer to work as the cocatalyst for proton reduction? (2) As Pt was used as the counter electrode in the system, Pt contamination should be taken into consideration.
4. In situ formed PTEB nanofiber on substrates can be directly used for photocatalytic hydrogen evolution in the presence and absence of sacrificial reagents. You can find similar experimental details in *J. Am. Chem. Soc.* 2017, 139, 1675-1683 and *Nat. Materials* 2016, 15, 611-615.
5. LSV curves, in a large electrochemical range, of PTEB nanofiber based photocathodes under both dark and light illumination should be provided.
6. Reviewer 2# has asked the author to provide monochromatic illumination quantum yield of this system. We quite agree with Reviewer 2# that this is a very important result to be provided. However, this result hasn't been given in the revised manuscript.

Reviewer #3 (Remarks to the Author):

The authors have carried out a very thorough revision of the manuscript. They performed additional experiments, added theory support and clarified the terminology. In my opinion the responses to all issues raised by all referees are satisfactory and have helped to improve the manuscript. I feel the paper is now ready for publication in *Nat Commun.*

We address the concerns of Reviewer 1# as follows:

Question 1:

Reviewers' comments: Reviewer #1 (Remarks to the Author): First of all, I would like to thank the authors' efforts to make this manuscript much better and most of the proposed questions have been clarified. However, following concerns are still not clear: 1. As H₂ gas, albeit very few, has been observed in the revised manuscript, the corresponding faradic efficiency of PEC H₂ evolution should be provided.

Response:

We thank the reviewer for the positive comments on the revised manuscript. Previously, we measured the gaseous product from the PEC cell using a gas chromatograph (GC) under light irradiation (100 mW cm⁻²), and a moderate amount of H₂ production (2.53 μmol in 10 h at 0 V vs. RHE) was detected on the PTEB photocathode. The amount of H₂ production is close to the value from theoretical calculation (Supplementary Fig. 24), suggesting the photocurrent of PTEB cathode mainly attributing to the PEC water reduction. Following the reviewer's suggestion, the Supplementary Fig. 24 was modified by adding the faradic efficiency and shown in Fig. R1. These results confirm that the photocurrent was indeed attributed to the PEC water reduction.

Figure R1 | Amount of evolved H₂ (black spots), recorded charge carrier (red line) and corresponding faradic efficiency on PTEB electrode in PEC cell with 0 V vs. RHE of applied bias in 0.01 M Na₂SO₄ under AM 1.5G irradiation (100 mW cm⁻²).

Question 2:

Although the authors have enumerated multiple examples of employing carbon-rich polymers for photo-induced reduction reactions, such as aryl aldehyde reduction, Cu^{II} reduction and pollutant degradation etc., the reduction of proton to hydrogen gas is a totally different reaction from the above-mentioned examples. As known to all, the reduction of proton to molecular hydrogen gas in general goes through the process of proton adsorption-reduction-hydrogen adsorption. Hence, the discussion of active sites for proton reduction by using this polymer is still not clear in my opinion. The authors must clearly clarify the true active sites for proton reduction.

Response:

We agree with the reviewer that it is interesting to identify the exact active sites for proton reduction in the PTEB network. However, it remains highly challenging so far to experimentally determine the exact active sites of polymeric catalysts using available characterization techniques (*Science* 2007, 317, 100; *Adv. Sci.* 2015, 2, 1500085), especially when there is no specific crystal face can be identified and controlled in the catalyst. Based on the reviewer's comments, we could simulate the reaction process of proton adsorption-reduction-hydrogen adsorption for H₂ production from water using density functional theory (DFT) calculations (*J. Am. Chem. Soc.* 2014, 136, 13629; *Energy Environ. Sci.* 2017, 10, 1186), and evaluate the free energy changes regarding to four different carbon atoms of PTEB. The reaction pathways for both single and dual sites H₂ evolution processes are shown in Fig. R2 a. The corresponding free energy variations for the two processes are shown in Fig. R2 b and c, respectively. The results indicate that site 1 and site 3 are favourable for single site hydrogen evolution, and the sites 1-2 are favourable for dual sites hydrogen evolution. The DFT calculation implies that carbon atoms of benzene ring (in PTEB) are dominate active sites for photocatalytic H₂ evolution, which agrees with the results from Cooper et al. (*J. Am. Chem. Soc.* 2015,

137, 3265; *Nature* 2015, 521, 41) proving that carbon-rich polymers (from phenylene and pyrene building blocks) are able to catalyse H₂ evolution from water in visible light. In addition, our UPS results (Fig. 3 in the main text) show that the band structures of PTEB are properly positioned to permit the transfer of electrons and holes, respectively, for water reduction. The results are now supported by DFT calculation of band structure as shown in Fig. R3, which indicates that the hydrogen evolution potentials at pH = 7 well located in the band gap of PTEB. All the energy levels are aligned according to vacuum level. Thus, the conduction band minimum (CBM) and valence band maximum (VBM) of PTEB are possible to catalyze H₂ evolution. The calculated band gap is 2.40 eV, which agrees with the experimental value of 2.51 eV.

The new results have been included in Fig. 6 in main text and Supplementary Figure 15, respectively.

Figure R2 | Density functional theory (DFT) calculations to investigate the H₂ evolution active sites. (a) Reaction cycles and active sites for single and dual sites H₂ evolution from water. **(b)** Free energy variations for

H₂ evolution *via* single site reaction pathway: 1, 2, 3, and 4 denote for different active sites as labelled in (a). (c) Free energy variations for H₂ evolution *via* dual sites reaction pathway: sites 1-2, 2-3, and 3-4 denote for different active sites as labelled in (a). The star (*) denotes catalyst (i.e. PTEB) surface.

Figure R3 | (a) Atomic structure of simulation supercell for PTEB. (b) High symmetry points in Brillouin zone. (c) Calculated band structure of PTEB by using vacuum level as reference.

Question 3:

In fact, bulk PTEB demonstrates negligible photocatalytic hydrogen evolution in the presence of sacrificial reagents. This is a very strange phenomenon! As shown in Figure 3d, PTEB acquires the ability for both proton reduction and even water oxidation. It can't reduce proton to hydrogen in the presence of sacrificial reagent, which is much easier to be oxidized than water. Why? Also, this result questions the true active sites of proton reduction in the PEC system. Two possibilities can be proposed: (1) do you think that residual Cu species in the polymer to work as the cocatalyst for proton reduction? (2) As Pt was used as the counter electrode in the system, Pt contamination should be taken into consideration.

Response:

It seems strange for us as well that bulk PTEB powder exhibited very poor photocatalytic performance for H₂ evolution in our preliminary test. However, it is also understandable if we consider that the only common point between the PTEB bulk powder and PTEB nanofibers lies in the same chemical structure. Other properties that are strongly related to the catalytic performance are different, such as morphology, dimension size, pore, surface area, molecular arrangement and electronic structure. These parameters will exert large effect on the whole processes of photocatalytic water reduction from reaction pathways, thermodynamics and kinetics, to charge separation and H₂ desorption. As shown in Fig. R4, the bulk PTEB powder synthesized in solution shows closely stacked microscopic particles, which is in contrast to the PTEB nanofibers (10-20 nm) that obtained by Cu-surface mediated approach. We would like to emphasize that this phenomenon is not unique to PTEB, since it has been a long-standing topic in understanding nanoscale effect in various catalysts, for example, the effects of particle size and morphology, and the catalytic properties of 2D materials, nanophases of carbon and nanoporous materials (*Nat. Mater.* 2015, 14, 505; *Nat. Commun.* 2015, 6, 6540; *Chem. Lett.* 2009, 38, 238). We finally would like to borrow the statement from X.H. Bao et al (*Natl. Sci. Rev.* 2015, 2, 183) to address the nanosize effect: "*Compared to bulk metals, nanoparticles (NPs) exhibit much larger total exposed surface areas and various combinations of surface structures, and electronic confinement effects within NPs may lead to major changes in the electronic structure. This raises the possibility of tuning the catalytic process.*"

Field Code Changed

We have shown that the PTEB nanofibers based photocathodes produced moderate amount of H₂ evolution (2.53 μmol) in 10 h reaction at 0 V vs. RHE (*i.e.* -0.6 V vs. Ag/AgCl) in PEC cell (100 mW cm⁻²). We appreciate that the reviewer offered us two very interesting papers (*J. Am. Chem. Soc.* 2017, 139, 1675 and *Nat. Mater.* 2016, 15, 611, refer to Question 4), so that we are able to know the appealing method to test directly the photocatalytic properties of PTEB nanofibers. In this process, a total amount of 11.4 μmol H₂ gas in the presence of sacrificial agent (*i.e.* triethanolamine) was

produced in 10 h reaction without noticeable deterioration of the activity within 30 h (Fig. R5, details refer to the Response to Question 4). Therefore, the average H₂ evolution rate of the PTEB nanofibers was about 1.14 μmol h⁻¹. If the mass weight (< 0.1 mg) of PTEB on the film was considered, an extremely high rate of > 11400 μmol h⁻¹ g⁻¹ for H₂ evolution could be obtained. Based on the above results, we strongly believe that the bulk PTEB powder could also exhibit excellent performance in photocatalytic water reduction, unless a suitable in-solution synthesis approach could be developed to achieve bulk PTEB with controlled nanoscale-structured morphologies. This is nevertheless beyond the focus of the current work.

Figure R4 | SEM images of (a) and (b) bulk PTEB powder synthesized by conventional in-solution method catalysed by CuBr, and (c) and (d) PTEB nanofibers synthesized by Cu-mediated interfacial method.

The active sites of PTEB for proton reduction were investigated by DFT calculation, which refers to the Response to Question 2.

1) After reaction, the PTEB samples were extensively washed by various solvents (pyridine, dichloromethane, 0.1 M HCl in methanol and methanol) to ensure there is no detectable Cu (by both XPS and EDX) in the film (Fig. 1g in main text, and Fig. S12 in Supplementary Information). And in

control experiments, we could not see any notable changes on the photocurrent before and after extensive washing.

2) Following the reviewer's suggestion, we have replaced the counter electrode from Pt to carbon and tested again the same sample. As shown in Fig. R5, there is no obvious difference in photocurrent between the two electrodes.

Figure R5 | Transient photocurrent density vs. time of PTEB photocathode using carbon as counter electrode at a bias of 0.3 V vs. RHE (*i.e.* -0.3 V vs. Ag/AgCl) under intermittent irradiation in 0.01 M Na₂SO₄.

Question 4:

In situ formed PTEB nanofiber on substrates can be directly used for photocatalytic hydrogen evolution in the presence and absence of sacrificial reagents. You can find similar experimental details in *J. Am. Chem. Soc.* 2017, 139, 1675-1683 and *Nat. Materials* 2016, 15, 611–615.

Response:

We thank the reviewer for the constructive comments and offering the two very interesting reference papers. Following the approach used in the two papers (*J. Am. Chem. Soc.* 2017, 139, 1675; *Nat. Mater.* 2016, 15, 611), we could test the photocatalytic activity of the PTEB nanofibers for H₂ evolution from water using a 25% triethanolamine aqueous solution as the sacrificial electron donor

under visible light irradiation ($\lambda > 420$ nm). As shown in Fig. R6, a total amount of 11.4 μmol H_2 gas was produced after 10 h reaction without noticeable deterioration of the activity within 30 h. The average H_2 evolution rate of the PTEB nanofibers was about 1.14 $\mu\text{mol h}^{-1}$. Furthermore, an extremely high rate of $> 11400 \mu\text{mol h}^{-1} \text{g}^{-1}$ for H_2 evolution could be obtained, when the mass weight (< 0.1 mg) of PTEB on the film was considered.

The new results have been included in Supplementary Figure 25.

Figure R6 | Cycle runs for the photocatalytic H_2 production from water over PTEB nanofibers under visible light irradiation ($\lambda > 420$ nm).

Photocatalysis experiments: the PTEB nanofibers sample ($3 \times 3 \text{ cm}^2$, *ca.* 230 nm) was placed at the center of the gas-closed reaction cell containing 120 mL 25% triethanolamine aqueous solution under magnetic stirring. The reaction temperature was kept at around 25 °C. A 200 W Xenon lamp equipped with a cutoff filter ($\lambda > 420$ nm) was applied to execute the photocatalytic reaction. The amount of H_2 produced was determined by gas chromatography equipped with a thermal conductivity detector (TCD).

Question 5:

LSV curves, in a large electrochemical range, of PTEB nanofiber based photocathodes under both dark and light illumination should be provided.

Response:

Following the reviewer's suggestion, new data were measured and shown in Fig. R7.

The new results have been included in Supplementary Figure 16.

Figure R7 | Linear scanning voltammetry (LSV) curves of PTEB photocathode under dark and simulated sunlight irradiation (100 mW cm^{-2}).

Question 6:

Reviewer 2# has asked the author to provide monochromatic illumination quantum yield of this system. We quite agree with Reviewer 2# that this is a very important result to be provided. However, this result hasn't been given in the revised manuscript.

Response:

We appreciate the valuable comments from the reviewer. We could not provide the monochromatic illumination quantum yield (MIQE) in the previous revised manuscript, because we found that the photocatalytic performance of bulk PTEB powder was too poor to gain solid data of MIQE. In addition, previously we did not know that the PTEB nanofibers film on substrate can be directly used for photocatalytic cell.

Following the method of the reference papers (refer to Question 4) that the reviewer suggested, in this revised version we were able to test directly the photocatalytic properties of PTEB nanofibers, and

found that the PTEB nanofibers exhibited a photocatalytic H₂ evolution rate of 1.14 μmol h⁻¹ (*i.e.* 11400 μmol h⁻¹ g⁻¹, refer to question 4). As such, the MIQE for H₂ evolution was measured using a similar experimental setup with a 420 nm band-pass filter. The MIQE was calculated based on the equation:

MIQE (%) = 100 × 2 × (the number of evolved H₂ molecules)/the number of incident photons. The number of the incident photons was determined using a radiant power energy meter (Newport). The produced H₂ molecules reached 1.3 μmol in 10 h, and thus the MIQE was calculated as:

$$\text{MIQE (\%)} = 100 \times 2 \times (1.3 \times 10^{-6} \times 6.022 \times 10^{23} \times 6.626 \times 10^{-34} \times 3 \times 10^8) / (9 \times 0.125 \times 10^{-3} \times 10 \times 60 \times 60 \times 420 \times 10^{-9}) = 1.83\%.$$

We address the concerns of Reviewer 3# as follows:

Comments:

Reviewer #3 (Remarks to the Author): The authors have carried out a very thorough revision of the manuscript. They performed additional experiments, added theory support and clarified the terminology. In my opinion the responses to all issues raised by all referees are satisfactory and have helped to improve the manuscript. I feel the paper is now ready for publication in Nat Commun.

Response

We greatly appreciate the reviewer for the positive comment.

REVIEWERS' COMMENTS:

Reviewer #1 (Remarks to the Author):

The authors have greatly improved the manuscript. I think it should be accepted for publication as it is.

Reviewer #3 (Remarks to the Author):

After the first round of revision, the other referees asked for further experimental and theoretical insights to clarify some remaining questions. The authors have sufficiently fulfilled these requests which helped to improve the manuscript further. Especially, the microscopic aspects regarding the active sites are quite interesting. In this context, it should be mentioned that the electronic band structure of 1,3,5-graphdiyne has already been reported in ref. 28, which should be cited. Overall, I feel the paper is now ready for publication in Nat Commun.